# Discrete Spatial Diffusion: Intensity-Preserving Diffusion Modeling

## Abstract

Generative diffusion models have achieved remarkable success in producing high-quality images. However, because these models typically operate in continuous intensity spaces—diffusing independently per pixel and color channel—they are fundamentally ill-suited for applications where quantities such as particle counts or material units are inherently discrete and governed by strict conservation laws like mass preservation, which limits their applicability in scientific workflows. To address this limitation, we propose Discrete Spatial Diffusion (DSD), a framework based on a continuous-time, discrete-state jump stochastic process that operates directly in discrete spatial domains while strictly preserving mass in both forward and reverse diffusion processes. By using spatial diffusion to achieve mass preservation, we introduce stochasticity naturally through a discrete formulation. We demonstrate the expressive flexibility of DSD by performing image synthesis, class conditioning, and image inpainting across widely-used image benchmarks, with the ability to condition on image intensity. Additionally, we highlight its applicability to domain-specific scientific data for materials microstructure, bridging the gap between diffusion models and mass-conditioned scientific applications.

## 1. Introduction

Diffusion-based generative models have emerged as powerful tools for high-quality image generation (Sohl-Dickstein et al., 2015; Ho et al., 2020; Song et al., 2021b). Typically, these models inject noise into the images, then learn to reverse this noise-adding process to recover meaningful structure. In most frameworks, this is based on an Itô Stochastic Differential Equation (SDE) with Gaussian noise. While effective for many vision tasks, these approaches inherently assume continuous pixel intensities, which can cause difficulty when dealing with the discrete nature of many datasets. Beyond vision-related tasks, such as in the physical sciences, there are many applications which require discrete physical quantities, such as particle counts in a sim-

ulation, or phases in materials microstructure. Conservation of total quantities (such as mass) can be critical for scientific applications, and so generative modeling which can operate under constrained, discrete pixel intensities would be scientifically valuable. Such a capability might also prove useful within vision tasks, such as inpainting and super-resolution.

Scientific and engineering studies of the natural world using computational techniques often involve discrete variables in space and/or time. On microscopic scales, everyday materials exhibit extremely complex structural patterns which encode the history of their formation, and play a large role in how the material functions on a macroscopic level. An important and wide-reaching field of study is materials microstructure, which is used in materials design (Gu et al., 2018), forensic analysis, hydrology (Blunt et al., 2013), energy storage (Simon & Gogotsi, 2008), and even medicine, such as in bone structure studies (Montoya et al., 2021). For example, crystal grain shapes can give rise to complex stress patterns which affect the yield strength of a metal (Calcagnotto et al., 2011). A materials microstructure can often be represented in terms of a small number of discrete phases which describe the underlying chemical structures involved in the microstructure. In sandstone, the overall arrangement of nanocrystals is highly disordered and gives rise to complex pore structures, through which subsurface water flows, and this microstructure can have an enormous influence on the rate of transport of fluids and contaminants. Microstructure of electrodes is also known to have an immense impact on the characteristics of electrochemical devices (Phogat et al., 2024). Small changes in thermodynamic properties can cause drastic changes in microstructure, such as in stainless steels (Xiong et al., 2010), necessitating study of microstructure as a function of phase contents. Furthermore, gathering real-world data on these systems is often complex and expensive; decades of work have been applied to computational modeling of the generation and consequences of microstructure prior to the widespread popularization of machine learning (Torquato, 2002).

In this work, we introduce Discrete Spatial Diffusion (DSD), a discrete-state Markov chain-based diffusion framework in which the forward process redistributes discrete units of intensity in space. Unlike previous diffusion models, DSD *exactly* preserves total intensity throughout both the forward

and reverse phases, ensuring that global properties—such as mass fractions or total particle count—are exactly conserved. We demonstrate that DSD not only enables scientific applications but also applies to more conventional tasks, like image generation and in-painting in discrete domains. By directly modeling discrete transitions, DSD paves the way for generative modeling under mass conservation, allowing models that specialize for constrained conditions in scientific applications and beyond.

## 2. Background

### 2.1. Related work

Among the body of literature on generative diffusion models, originating from the pioneering work of Sohl-Dickstein et al. (2015), the most relevant to our work fall into two broad categories: (1) those employing discrete-state Markov chains to introduce noise in the forward process (Hoogeboom et al., 2021; Austin et al., 2021; Campbell et al., 2022; Santos et al., 2023; Sun et al., 2022; Lou et al., 2024), and (2) those incorporating spatial dynamics into the forward diffusion process (Bansal et al., 2022; Rissanen et al., 2022; Hoogeboom et al., 2021).

Generative diffusion modeling based on discrete-state Markov chains has become an active area of research in recent years. Early work, such as Hoogeboom et al. (2021); Austin et al. (2021), introduced discrete-state and discrete-time Markov chains as an alternative to the Gaussian noise used in conventional diffusion models (Sohl-Dickstein et al., 2015; Ho et al., 2020; Song et al., 2021b). Campbell et al. (2022) generalized these formulations to a continuous-time framework, providing a more rigorous theoretical foundation for discrete-state generative diffusion modeling. Santos et al. (2023) employed operator algebraic analysis to formally establish the existence of the reverse-time dynamics and derived the stochastic generator for arbitrary discrete-state Markov processes. Similar formulations were independently developed by Sun et al. (2022) and Lou et al. (2024), with an emphasis on defining and estimating score functions for discrete-state systems. The Markov process operates in intensity space in all the aforementioned diffusion models, treating each pixel as an independent stochastic process (Fig. 1(a): Gaussian; Fig. 1(b): Discrete).

This study focuses on a spatially correlated process for generative modeling for two reasons: (1) for structured images, a more natural approach is to incorporate spatial correlations into the generative process, and (2) spatially decorrelated noise makes it difficult to preserve total intensity. A spatially correlated approach has been explored for continuous systems. Cold Diffusion (Bansal et al., 2022) introduced a deterministic blurring transformation, where image degradation follows a predefined forward process,

and reconstruction is learned as an inverse mapping. However, lacking a probabilistic latent distribution (as in VAEs (Kingma & Welling, 2014)), Cold Diffusion is not a true generative model. Inverse Heat Dissipation Model (IHDM, Rissanen et al. (2022)) uses the heat equation as a corruption model. Since the heat equation is deterministic and reversible (except for the homogeneous solution at $t \to \infty$ is singular), a naïve inversion would again result in deterministic reconstructions. Uncorrelated Gaussian noise was added to the heat equation to overcome this limitation, relaxing the deterministic process into a probabilistic Itô diffusion. Later, Blurring Diffusion Model (BDM, Hoogeboom & Salimans (2022)) recognized that IHDM could be recast as a Gaussian diffusion model in the spectral domain. BDA extended IHDM and achieved SOTA generative performance, validating the hypothesis that spatially structured diffusion processes can enhance image generation. Nevertheless, the probabilistic formulation of IHD and BDM only preserves mass on average, not exactly per-sample, and their continuous-state nature makes it difficult to apply to discrete datasets.

Our goal of generating samples with exactly conditioned total intensity aligns with conditional diffusion modeling. However, existing approaches all rely on some degree of approximation. Song & Ermon (2019) proposed a simple conditional sampling method by passing class labels into the neural network during training, but this does not guarantee exact enforcement of the condition in generated samples. A more structured approach was introduced by Chung & Ye (2022); Chung et al. (2022c;b), which interleaved projection steps with diffusion sampling to enforce linear constraints in image generation. However, these projections disrupt the exactness of the forward corruption and reverse inference dynamics (Anderson, 1982; Campbell et al., 2022; Santos et al., 2023), leading to a mismatch between the projected and true data manifolds. To address this, Chung et al. (2022a) eliminated projection steps but instead relaxed deterministic constraints into a probabilistic formulation via a noisy measurement model. However, this method does not apply to deterministic constraints, as it becomes singular in the limit of zero measurement noise. An alternative approach leverages Bayes' theorem for a posteriori conditional sampling, that is, $p(S|C) \propto p(S)p(C|S)$, where "S" stands for samples and "C" for condition(s). Because $p(S)$ is given by a trained unconditional diffusion model, conditioning can be performed if one has $p(C|S)$, which is however intractable for arbitrary data distributions[1]. Existing methods approximate this term crudely or by training a separate classifier as in Song et al. (2021b), or by a Gaussian approximation with moment-matching as in (Finzi et al., 2023;

---

[1]It is challenging because the constraint is imposed on the *final samples at the end of the inference*, but the conditioning "S" are *samples generated during the inference*.

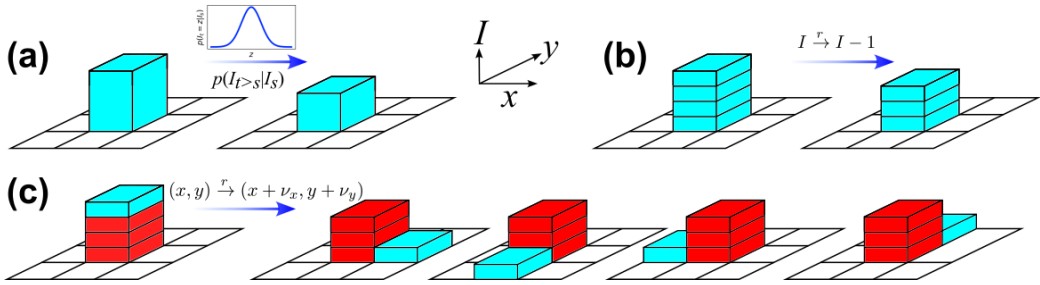

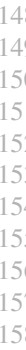

*Figure 1.* Schematic diagrams of various diffusion models. (a) Gaussian Diffusion relies on the Ornstein–Uhlenbeck diffusion process in the intensity space.(b) Previous discrete-state diffusion models rely a discrete-state Markov process of the transition of discretized intensities. (c) Discrete Spatial Diffusion (this study) relies on a Markov jump process of the intensity units on the discrete lattice, conserving the total intensity seperately in each color channel.

Du et al., 2024). None of these methods guarantees the generated samples are exactly conditioned.

### 2.2. Summary of our contributions

The principal contribution of our work is that it provides a new capability for diffusion models to preserve intensity exactly in a fully discrete-state context. The approach is based entirely on how the diffusion process is built, and how the model is trained; it is readily usable with existing diffusion model neural network (NN) architectures. To our knowledge, this is the first diffusion model to incorporate a spatially correlated noise, which is accomplished using a stochastic jump process allow units of intensity to perform a random walk. This fact also demonstrates that more complex noise processes can themselves be tractable. We furthermore demonstrate that such a model is powerful enough for conventional image synthesis tasks. The relevance and power of the approach is then demonstrated through application to scientific data in the field of materials microstructure, where the ability to generate complex data-driven images with constrained total intensity is highly desirable.

## 3. Methods

### 3.1. Corruption Process

In this manuscript, we adopt the language of image processing and consider 2-dimensional images, although the context and spatial dimensionality of the data are not constrained by the mathematical framework provided here. We treat a digital image with discretized intensities $I_{x,y,c} \in \mathbb{Z}_{\geq 0}$ at pixel $(x, y) \in \{1, \ldots, W\} \times \{1, \ldots, H\}$ in color channel $c \in \{1, \ldots, C\}$. Within the DSD framework, the image is treated as a spatially organized collection of particles; one for each intensity unit. Below, we will interchangeably use "particles" and "intensity units" to denote these fundamentally discrete units. Specifically, $I_{x,y,c} = n$ implies $n$ particles of type $c$ at location (x,y), and the total number

of particles of the system is $\sum_{x=1}^{H} \sum_{y=1}^{W} \sum_{c=1}^{C} I_{x,y,c}$. In the forward stochastic process with the time parameter $t$, each of the particles in the system *independently* performs a continuous-time and discrete-state random walk:

$$(x, y, c) \xrightarrow{r} (x + \nu_x, y + \nu_y, c), \tag{1a}$$

$$\nu := (\nu_x, \nu_y) \in \{(1,0), (-1,0), (0,1), (0,-1)\} \tag{1b}$$

where $r$ is the transition rate of the particle jumping to one of their nearest neighbors, and $\nu$ is a set of four directions the particles can hop to their nearest neighbors. Note that the particles perform jumps in the $(x, y)$ space at random times, but do not change their color coordinate $c$. A schematic diagram is shown in Fig. 1c. We impose either no-flux boundary condition, such that the transition rates of jumping out of the image domain are zeros, or periodic boundary conditions so that a jump to $x = W + 1$ becomes a jump to $x = 0$, vice-versa, and analogously for $y$. Note that the forward process conserves the total number of particles, $\sum_{x=1}^{H} \sum_{y=1}^{W} I_{x,y,z}$ in each color channel independently.

We refer to the spatial hopping process (1) as the *Discrete Spatial Diffusion* (DSD), noting the "discreteness" refers to both the discretized intensity units and the discreteness of the spatial lattice $\{1, \ldots, W\} \times \{1, \ldots, H\}$ where the particles are allowed to reside. DSD, as well as similar discrete-state random walks, have been extensively studied in non-equilibrium statistical physics and stochastic processes (Van Kampen (2007), Gardiner (2009), Giuggioli (2020) and references therein). The evolution of the probability distribution of the single random walk in the continuum space limit, under the appropriate scaling of the transition rate (Einstein, 1905), converges to the Fokker–Planck Equation (FPE, Van Kampen (2007); Risken (1984)), which is mathematically identical to the heat equation. Because of the duality between the probabilistic FPE and the deterministic heat equation (Lawler, 2010), DSD can be considered as a microscopic description of the macroscopic heat dissipation that inspires IHD and BDM. Notably, the correlated noise is built in DSD, in contrast to the heuristic addition

of uncorrelated Gaussian noise in IHD and BDM. Figure 2 illustrates the application of DSD to a sample image. Due to the stochasticity of the random jumps, the limiting behavior ($t \to \infty$) of this process is a random configuration with no discernible structure or similarity to the original spatial organization aside from the conserved global particle counts in each color channel.

We use $(X_t, Y_t, C_t)$ to denote the random process in $(x, y, c)$ space, and $(x_0, y_0, c_0)$ are the initial condition of a specific particle. We use $I_t$ to denote the randomly corrupted image at the time $t$, where $[I_t]_{x,y,c}$ is the total number of particles at $(x, y)$ in color channel $c$. The process can be represented in these two dual representations: with $(X_t, Y_t, C_t)$ the process is formulated in the frame of a moving particle (the Lagrangian frame), and with $I_t$ the process is formulated as a histogram in space-time (the Eulerian frame). Below, we will use these two representations interchangeably.

The forward solution, the transition probabilities $p_t(x, y, c | x_0, y_0, c_0) := \mathbb{P}\{X_t = x, Y_t = y, Z_t = z | X_0 = x_0, Y_0 = c_0, C_0 = c_0\}$, can be computed by integrating the Master Equation (Van Kampen, 2007; Gardiner, 2009; Weber & Frey, 2017). This corresponds to exponentiating the Markov transition matrix of the process defined in Eq. (1). While the matrix exponential required numerically for no-flux boundaries is expensive, the solution can be stored and reused to generate corrupted images and to compute the reverse-transition rates (see Sec. 3.3) for learning. When periodic boundary conditions are imposed, the transition matrix is diagonal in the discrete Fourier space, facilitating the efficient computation of $p_t(\cdot|\cdot)$.

### 3.2. Designing Noise Schedules by Structural Similarity Index Metric (SSIM)

The corruption process (1) is a time-homogeneous stochastic process. Consequently, the noise induced in the system, per particle and per unit time, remains constant. However, it has been shown that inhomogeneous noise schedules can facilitate learning (Nichol & Dhariwal, 2021). We use the formulation of a recent study (Santos & Lin, 2023) identified the unique correspondence between non-uniform observation times in a homogeneous Ornstein–Uhlenbeck process (Uhlenbeck & Ornstein, 1930) and noise schedule in conventional diffusion models (Ho et al., 2020; Song et al., 2021a). We follow the same philosophy as Santos & Lin (2023) to construct a sequence of observation times $t_0 = 0 < t_1 < t_2 < \ldots < t_T = 1$, at which we will generate random samples for learning. Here, $T$ is the total number of discrete times we will generate corrupted sample images for learning.

We adopt a heuristic approach to construct the discrete times. The idea is to use a metric to quantify how much the "quality" of the images has been degraded up to time $t$, and we aim

to design $t_k$'s such that the metric degrades from $k = 0$ to $k = T$ as evenly as possible. In this manuscript, we chose the Structural Similarity Index Metric (SSIM, Wang et al. (2004)) between the corrupted image and the original one. We generalize a generic monotonic relation between $k$ to $t_k$ proposed by Santos et al. (2023):

$$\Phi\left(e^{-\tau_2 t_k}\right) \triangleq \frac{(k-1)\,\Phi\left(e^{-\tau_2}\right) - (T-k)\,\Phi\left(e^{-\tau_1}\right)}{T-1}, \quad (2)$$

where $\Phi(p) := \log p/(1-p)$ is the logit function, $\tau_1$ and $\tau_2$ are parameters used to construct the observation times. Note that $t_T = 1$ in the above parametrization. Specifically, we tune $\tau_1$, $\tau_2$ and the unit transition rate $r$ in process (1), using a subset of training samples, aiming to cover an even degradation of the SSIM throughout observation times. We found that setting $\tau_1 = 7.5$ and $\tau_2 = 2.5$, and $r = 120\text{-}160$ is sufficient for numerical experiments. We remark that the choice of the functional form in Eq. (2) is arbitrary and without any theoretical foundation; we only treat Eq. (2) as a versatile monotonic fitting function, whose corresponding SSIM degradation is empirically more symmetric than polynomial and cosine schedules (Nichol & Dhariwal, 2021) for the DSD process (see Appendix Fig. 6).

### 3.3. Reverse-time process

Following the general theoretical framework developed in (Campbell et al., 2022; Santos et al., 2023), there exists a reverse-time process that evolves in opposite time and whose joint probability distribution is identical to that of the forward process (1). Specifically, the reverse-time process corresponding to process (1) is

$$(x, y, c) \xrightarrow{r\,\frac{p_t(x+\bar{\nu}_x, y+\bar{\nu}_y, c|x_0, y_0, c_0)}{p_t(x, y, c|x_0, y_0, c_0)}} (x + \bar{\nu}_x, y + \bar{\nu}_v, c), \quad (3)$$

where the admissible reverse-time transitions $\bar{\nu} = (\bar{\nu}_x, \bar{\nu}_y) := \in \{(-1, 0), (1, 0), (0, -1), (0, 1)\}$ are the reversed direction of the forward jumps ($\bar{\nu}_x = -\nu_x$, $\bar{\nu}_y = -\nu_y$). The framework ensures the same boundary condition to be imposed (no-flux or periodic, according to the forward process). We note that the reverse-time process, and therefore the generated images, also conserve the total particle number per color channel.

We note that the reverse transition rate depends on both the initial condition $(x_0, y_0, c_0)$ of a particle and the forward solution $p_t(x, y, c|x_0, y_0, c_0)$, $\forall(x, y, c)$. This is analogous to conventional diffusion models, where either the reverse-time drift (Sohl-Dickstein et al., 2015; Ho et al., 2020) or the score function (Song et al., 2021b) formally depend on the initial sample and the solution of the forward process. However, during the inference, the initial particle configuration is not known, and as such, we train an NN to learn the reverse transition rates using samples $I_t$ generated from

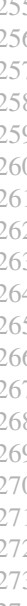

Figure 2. (a) The forward processes for Gaussian Diffusion (Ho et al., 2020), Inverse Heat Dissipation Model (Rissanen et al., 2022), and Discrete Spatial Diffusion (ours) applied on an image, sampled at discrete times. (b) Percentage change in mass relative to the original image under the forward process.

the forward process (1) at $t > 0$. Additionally, the particles are indistinguishable, but the rate prescribed in Eq. 3 is *per-particle*, raising the question: what is the appropriate *per-pixel* reverse transition rate that the NN ought to model? This question can be answered by performing the survival analysis of the many-particle system in light of the independence of particle motion; see Appendix A for a derivation. Intuitively, this can also be derived by combining the first-reaction method (Gillespie, 1976) and inhomogeneous Poisson process (e.g., see Corbella et al. (2022)). The analysis shows that the reverse transition rate of the first jump of $n = [I_t]_{x,y,c}$ particles is simply the sum of the instantaneous transition rates:

$$\bar{r}_{\bar{\nu},x,y,c} = r \sum_{i=1}^{n} \frac{p_t(x + \bar{\nu}_x, y + \bar{\nu}_y, c | x_0^{[i]}, y_0^{[i]}, c_0^{[i]})}{p_t(x, y, c | x_0^{[i]}, y_0^{[i]}, c_0^{[i]})}, \quad (4)$$

The above prescribes the rate for *the first* of all the particles (which is $[I_t]_{x,y,c}$) to jump to one of its neighboring pixels. It also prescribes the rate that the NN will model. This rate is still time-dependent through the dependence on the forward solution $p_t$, similar to standard continuous-time diffusion models.

### 3.4. Loss functions

Our goal is to provide the corrupted images $I_t$ at a sampled time $t > 0$ to a neural network (NN) and to train it to predict the reverse-time transition rates (3). We denote the NN modeled rates as $r_{\bar{\nu},x,y,c}^{\text{NN}}(I_t, t) \in \mathbb{R}_+^{4 \times H \times W \times C}$. The four in the first dimension here corresponds rates for four nearest-neighbor transitions.

There exist two approaches to formulate the loss functions. The first and more common approach adopts a metric and

heuristically matches the NN prediction and the ground truth. DDPM (Ho et al., 2020), score-matching (Song et al., 2021b), and flow-matching (Lipman et al., 2022). When predicting rates, we extend these schemes to "rate-matching", where we minimize the chosen norm of the difference between the predicted and true rates: $\bar{r}^{\text{NN}}$ and $\bar{r}$. For example, for using L1, a loss $\mathcal{L}$:

$$\mathcal{L}_{\text{L1}} = \mathbb{E}_{I_{t_k}, k} \left[ \text{mean}(\left| \bar{r}^{\text{NN}} - \bar{r} \log \bar{r}^{\text{NN}} \right|) \right]. \quad (5)$$

Here, $k \in \{1, \dots T\}$ is uniformly sampled, $I_{t_k}$ is drawn from the random process (1) at the sampled times, $\bar{r} = \bar{r}_{\bar{\nu},x,y,c}(I_t, t | I_0)$ is the theoretically computed reverse-time transition rate (4), $\bar{r}^{\text{NN}} = \bar{r}_{\bar{\nu},x,y,c}^{\text{NN}}(I_t, t)$ is the NN-predicted reverse-time transition rate, and the mean is over all the indices $(\bar{\nu}, x, y, c)$. The second and more principled approach is through minimization of the negative log-likelihood $L$ of the NN-induced process to predict the analytical reverse-time process (Sohl-Dickstein et al., 2015; Campbell et al., 2022; Santos et al., 2023):

$$\mathcal{L}_L = -\log L = -\mathbb{E}_{I_t} \left[ \int_0^\infty \sum \left( \bar{r}^{\text{NN}} - \bar{r} \log \bar{r}^{\text{NN}} \right) \mathrm{d}t \right]. \quad (6)$$

Because we only observe the process at discrete times prescribed in Eq. (2), we approximate the continuous-time integration above by

$$\log L = \mathbb{E}_{I_{t_k}, k} \left[ (t_k - t_{k-1}) \sum \left( \bar{r}^{\text{NN}} - \bar{r} \log \bar{r}^{\text{NN}} \right) \right], \quad (7)$$

where we again take expectation over randomly sampled $t_k$ and $I_k$. In this study, we experimented with both loss functions and did not discover any noticeable difference, giving evidence that the DSD forward process (1) is not

sensitive to the choice of the loss function. This is favorable over the Gaussian diffusion models as discussed in Ho et al. (2020), which used the heuristic approach to improve over Sohl-Dickstein et al. (2015), which adopted the second approach. We focus on learning the transition rates of the reverse-time dynamics, which is distinct from the ratio-matching approach (Sun et al., 2022; Lou et al., 2024) which focuses on learning the probability distribution $p_t(\cdot)$, although a similar formulation ("implicit score entropy") proposed by (Lou et al., 2024) can be regarded as the process likelihood (7) first proposed in Santos et al. (2023). Algorithm 1 describes the DSD training pseudocode.

### 3.5. Sampling with Adaptive Time Stepping

Once trained, the neural network will predict reverse-time rates (4), given the configuration of system, $I_t$, at time $t \geq 0$. Because the reverse rates are time-dependent, we could generate the exact sample paths of the inhomogeneous Poisson process by integrating the survival function of the first reaction on each pixel in each color channel (see e.g., algorithms reported in Corbella et al. (2022)). However, this approach is not computationally efficient, so we resort to $\tau$-leaping (Gillespie, 2001), an integrator that has been adopted by essentially all continuous-time and discrete-state diffusion models (Campbell et al., 2022; Santos et al., 2023; Winkler et al., 2024; Ren et al., 2024), analogous to the Euler's method for ordinary or partial differential equations and Euler–Maruyama for Itô SDEs. The central idea of $\tau$-leaping is to approximate the reverse-time transition rates $\bar{r}$ as a fixed constant in a small enough window $(s - \tau, s)$, assuming the time-dependent rates change slowly in the period, a condition often termed as the "leap condition" (Gillespie, 2001; Cao et al., 2005). With this assumption, the original $\tau$-leaping algorithm by Gillespie (2001) generates Poisson random numbers to update the system's discrete states. However, this approach could sometimes lead to a negative population of particles, which cannot happen in the process, due to violations of the leap condition. Mitigation strategies exist (for example, see Gillespie & Petzold (2003), Cao et al. (2005) and Cao et al. (2006)), however, some of them are limited to small reaction networks and not suitable for the DSD sampling task, which involves a very large number of $(4 \times H \times W \times C)$ of transition rates to estimate.

As such, we propose a more efficient (but arguably cruder) approach to select the stepper $\tau$ adaptively. Our idea is to combine the binomial $\tau$-leaping (Tian & Burrage, 2004; Chatterjee et al., 2005) and the Courant–Friedrichs–Lewy (CFL) condition (Courant et al., 1928) to conservatively determine the adaptive step size $\tau$. Specifically, since the jump scale is fixed at the pixel length scale, the timescale $\tau$ fully determines the CFL condition. The idea is to choose a $\tau$

such that the CFL number is fixed throughout the inference[2]. To achieve this, we compute the reverse-time transition rates $\bar{r}_{\bar{\nu}}$ for each pixel in each channel, noting that the probability of a particle in that channel will jump to one of its neighboring locations is $\bar{r}_{\bar{\nu}}\tau$. Then, we determine $\tau$ by fixing the largest probability across all the pixels and color channels at a constant. Algorithm 2 describes the DSD inference pseudocode.

## 4. Computational Experiments

We employ the Noise Conditional Score Network (NCSN++) (Song et al., 2021b) with two modifications: the final convolutional layer outputs 4 times the number of input channels (e.g., 3 for RGB) to represent four directions (up, down, left, right), and we use a SoftPlus activation function to ensure non-negativity in the predicted rates. The hyperparamters can be found in Appendix D.

### 4.1. Image synthesis benchmarks

While the primary motivation for developing DSD is to enable generative modeling under a strict intensity constraint, we first demonstrate the approach on MNIST (LeCun et al., 2010) and CelebA (Liu et al., 2015), demonstrating that the approach can achieve reasonable generative performance for these commonly studied datasets. Unconditionally generated samples are shown in Fig. 3 (a) and (b). Generated samples of the CelebA dataset show that complex patterns including human facial features, lighting, and textures can be captured by DSD.

Next, we explored additional applications of mass conservation for handwritten digits. In Fig. 3 (c), we show results from an in-painting experiment with a fixed mask. In this training, the no-flux boundary conditions were implemented inside the image region, and particles outside of this region were fixed. Throughout the generation process, the disordered particles inside the mask align themselves given the structure outside of the mask. Given the same structure outside of the masked region, we varied the number of particles in the active region, leading to the generation of different digits, as exemplified in the Fig. 3 (c). Additionally, we trained a conditional DSD model that employed the standard class-conditioning (Song et al., 2021b). Figure 3 (d) illustrates the class-conditioned generated images with dif-

---

[2]Even though CFL condition is more commonly used in PDE integrators, the concept can be applied for our stochastic system. Suppose the reverse-time rate is $\bar{r}$. On average, the particle would move at a timescale $1/\bar{r}$ to one of its neighbors, traveling $\Delta x$. Then, the velocity $c = r\Delta x$. The CFL condition is then $c\Delta t/\Delta x$ where in our scheme $\Delta t$ is the $\tau$; thus, the classical CFL convergence condition translates to the obvious bound of transition probability $\bar{r}\tau < 1$. This motivates us to ensure a conservative estimation of $\tau$, but enforcing a small $\bar{r}\tau$ to reduce the error.

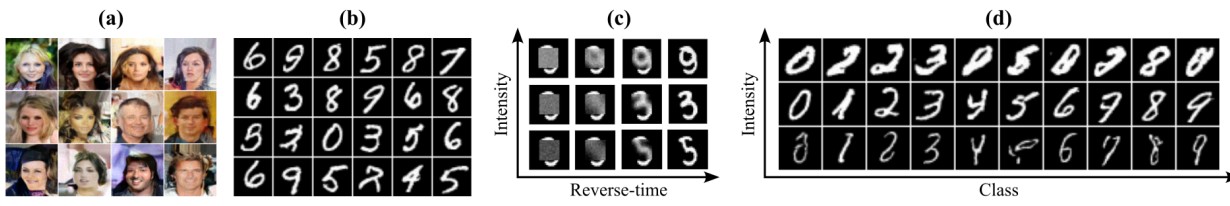

*Figure 3.* (a) Unconditional CelebA generation. (b) Unconditional MNIST generation. (c) Unconditional inpainting on MNIST; 15% difference of conditioning intensity between consecutive rows. (d) Conditioned MNIST generations across different intensities.

ferent total numbers of particles, varying from low, typical, and high total intensities. While these do exhibit some artifacts, DSD surprisingly learns the spatial structure of the digits and generates "Bolder" or "Lighter" digits without saturating the upper bound of the intensity (i.e. 255 for `uint8`). This would not have been precisely realizable using conventional diffusion models.

### 4.2. Subsurface rock microstructures

The microstructure of subsurface rocks governs a wide range of physical processes, including fluid transport, electrical resistivity, and mechanical deformation (Blunt et al., 2013). This originates from connected pores on the nano- and micro-scale, which vary in size, structure, and coordination degree across rock types. High-resolution 3D imaging via X-ray microtomography enables detailed pore-scale reconstructions, but these scans are expensive and limited to sample sizes on the order of millimeters to centimeters (Cnudde & Boone, 2013). While direct imaging of rock microstructure is costly, measuring porosity (defined here as average intensity over the image) across large formations is inexpensive and can be performed without specialized equipment (Leonard, 1948; Passey et al., 1990). This enables large-scale field measurements of porosity, even when high-resolution microstructural data is unavailable.

To overcome this limitation, synthetic models are frequently used to generate representative pore structures for computational physics studies (Øren & Bakke, 2002). However, conventional reconstruction techniques impose strong geometric assumptions that fail to capture the heterogeneity observed in real rocks like Berea Sandstone, Savonnières Carbonate, and Massangis Carbonate. We trained DSD models using these types of rock samples, which represent a broad spectrum of pore structures (including granular, fossiliferous, and dissolution-driven features) across two lithologies: sandstone and carbonate. A description of the training datasets is provided in Appendix E. Figure 4 presents representative outputs from our models trained on $256 \times 256$ binary images. The generated samples successfully replicate key statistical properties of the original datasets, including spatial correlation and pore size distribution, both of which are critical for fluid transport. Given that DSD allows for precise control over total porosity, one can generate syn-

thetic microstructures that match the porosity measured in the field, enabling the reconstruction of representative pore-scale samples even in the absence of direct imaging. The model accurately reconstructs microstructural statistics relevant to flow in the subsurface–for details see Appendix E.1.

### 4.3. Lithium-ion electrodes

Electrodes in lithium-ion batteries are porous materials with a complex microstructure that governs key properties like ion transport and electrochemical performance. Nickel-manganese-cobalt cathodes, among the most common, are composed of three phases: the active material driving the electrochemical reaction, the carbon binder ensuring electrical conductivity and mechanical stability, and the pore space filled with electrolytes. The active material is expensive, creating a strong economic incentive to understand how its volume fraction and distribution influence electrode behavior. While tomographies are needed for studying microstructures and enabling computational modeling, acquiring diverse datasets is challenging (Deng et al., 2021). To overcome this, researchers often rely on computational methods to generate synthetic microstructures (Duquesnoy et al., 2023). While generative adversarial networks have been explored for this purpose, they did not control phase volume ratio parameters (Gayon-Lombardo et al., 2020). We trained a DSD model on tomography data (Usseglio-Viretta et al., 2018), where two color channels were used to represent the carbon binder and active materials. The results, shown in Fig. 5, demonstrate DSD enables precise tuning of phase volume fractions, making a powerful tool for systematically studying and optimizing electrode microstructures. For more details on datasets and reconstruction metrics applied to these samples, see Appendix F.

## 5. Limitations

The computational cost of forward sampling during training and reverse-time sampling during inference in DSD scales linearly with the *total intensity* of the image. While this makes DSD highly efficient for low-bit-depth or binary datasets, it may become less efficient than other techniques for higher-resolution images or datasets with higher intensity saturation, such as standard *uint8* images. Addi-

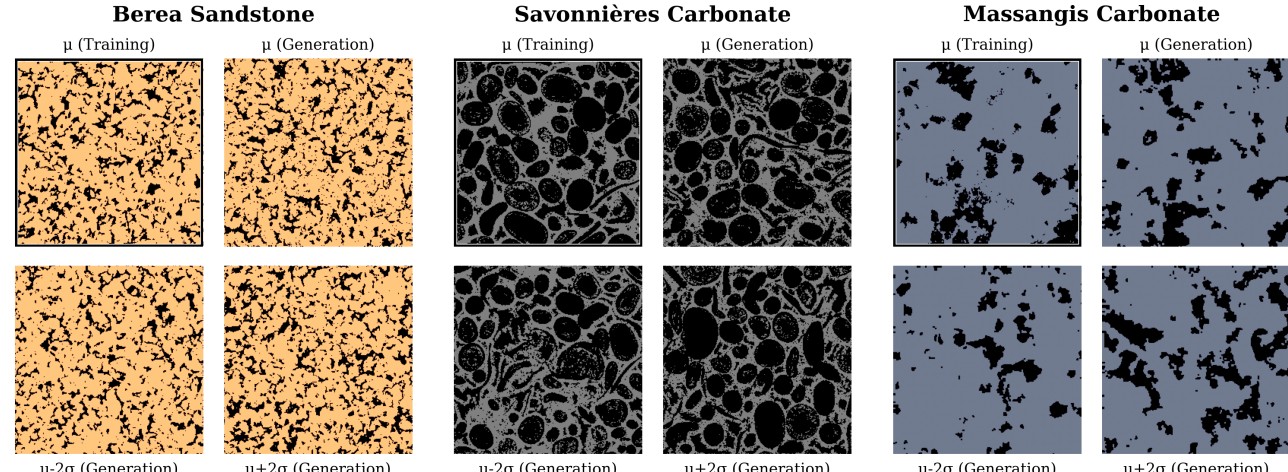

**Berea Sandstone**  **Savonnières Carbonate**  **Massangis Carbonate**

μ (Training) μ (Generation) μ (Training) μ (Generation) μ (Training) μ (Generation)

μ-2σ (Generation) μ+2σ (Generation) μ-2σ (Generation) μ+2σ (Generation) μ-2σ (Generation) μ+2σ (Generation)

*Figure 4.* Schematic representation of three rock types: Berea Sandstone, Savonnières Carbonate, and Massangis Carbonate. The first image (top left) for each rock type shows one training sample, while the second one (top right) displays the generated sample conditioned on the mean intensity $\mu$ of the training set. The third (bottom left) and fourth (bottom right) samples illustrate the generated samples conditioned on $\mu - 2\sigma$ and $\mu + 2\sigma$, respectively, where $\sigma$ represents the standard deviation of the training set intensity distribution.

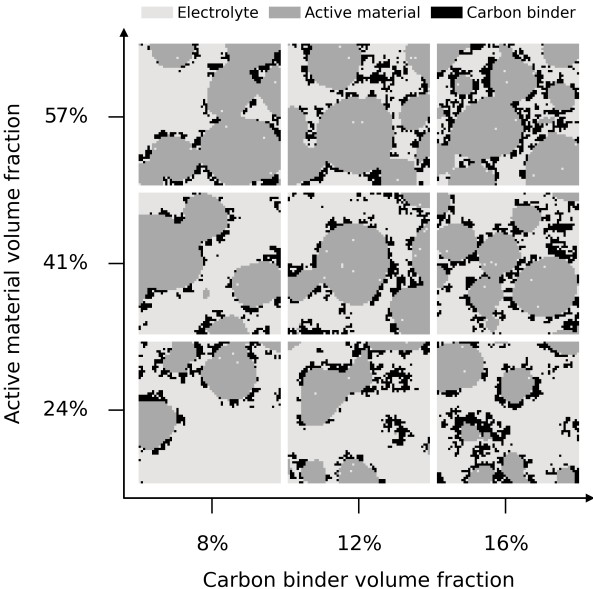

*Figure 5.* Generated cathodes with various exactly conditioned phase volume fractions. Carbon binder domain in black, active material particles in gray, electrolyte in white.

tionally, enforcing strict intensity conservation requires a custom forward process code (Eq. (1)) and a novel sampling scheme, deviating from conventional Gaussian diffusion models. This introduces a steeper learning curve for practitioners accustomed to standard diffusion approaches. However, we argue that these trade-offs are necessary to achieve exact constraint enforcement, which is not possible with existing methods.

## 6. Conclusion

We introduced Discrete Spatial Diffusion (DSD), a fully discrete, mass-preserving generative model approach for images and scientific data. The foundation is to use discrete-state, continuous time statistical processes incorporating jump dynamics, rather than SDEs, as a foundation, and in particular is the first discrete diffusion model to explore spatially correlated noisification. DSD demonstrates competitive quality on standard benchmarks while enabling exact global constraints in total intensity (particle count, or mass) that are critical in many scientific applications. By preserving these constraints in both forward and reverse processes, DSD provides for exactly constrained data generation, which we explored on image synthesis and domain-specific datasets. It also demonstrates that more complex statistical processes (in this case, random walks) can be used for diffusion modeling, perhaps opening the door for further models to exploit structure in their dynamics such as conservation laws and symmetries.

## Impact Statement

This paper presents work whose goal is to advance the field of generative modeling for spatial data. There are many potential societal consequences for generative modeling research, however, these are largely unspecific to the research presented here. This paper advances the ability to generate image data under constraints, which could improve the capabilities of generative models, and in particular in settings which are treated mathematically, such as physical simulations; we believe this does not broaden the scope of ethical concerns associated with generative models.

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

## A. Deriving reverse-time transition rates

Here, we derive the reverse-time transition rate. Because the particles are moving independently, it is sufficient to discuss $n$ particles colocalized at $(x, y)$ in channel $c$, and the conclusion applies to other locations and color channels. For brevity, we will drop the $(x, y, c)$ dependence in this section when the context is clear. Let us index the particles by $i = 1 \ldots n = [I_t]_{x,y,c}$. For each of the $n$ particles, the reverse-time transition rates moving to $(x + \bar{\nu}_x, y + \bar{\nu}_y)$, where $(\bar{\nu}_x, \bar{\nu}_y) \in \{(-1, 0), (1, 0), (0, -1), (0, 1)\}$ is

$$\bar{r}_{\bar{\nu}}^{[i]}(t) = r \frac{p_t \left( x + \bar{\nu}_x, y + \bar{\nu}_y, c | x_0^{[i]}, y_0^{[i]}, c_0^{[i]} \right)}{p_t \left( x, y, c | x_0^{[i]}, y_0^{[i]}, c_0^{[i]} \right)}, \quad (8)$$

according to the general theory of reverse-time dynamics for continuous-time Markov systems (Campbell et al., 2022; Santos et al., 2023). We now perform the survival analysis for the inhomogeneous process. Within time $dt$, the probability that particle $i$ leaves $(x, y, c)$ and moves to $(x + \bar{\nu}_x, y + \bar{\nu}_y, c)$ is $\bar{r}_{\bar{\nu}}^i(t) \, dt + \mathcal{O}(dt^2)$. As such, the probability of the particle remains at $(x, y, c)$ at time $t - dt$ is $1 - \sum_{\bar{\nu}} \bar{r}_{\bar{\nu}}^{[i]}(t) \, dt + \mathcal{O}(dt^2)$. Thanks to the independence between the particle dynamics, the probability of *all* $n$ particles remaining at $(x, y, c)$ at time $t - dt$ (recall that we are evolving the reverse-time dynamics) is $1 - \sum_{i=1}^{n} \sum_{\bar{\nu}} \bar{r}_{\bar{\nu}}^{[i]}(t) \, dt + \mathcal{O}(dt^2)$. Then, the probability of *no* particle leaving at a previous time $t - \Delta t$, where $\Delta t := N dt$ is

$$\prod_{k=1}^{N} \left[ 1 - \sum_{i=1}^{n} \sum_{\bar{\nu}} \bar{r}_{\bar{\nu}}^{[i]} \left( t - (k - 1) \, dt \right) dt \right] + \mathcal{O}(dt^2), \quad (9)$$

which by sending $dt \downarrow 0$ leads to the continuous-time survival function:

$$\mathbb{P}\{\mathcal{T} > t\} = \exp\left[ -\int_0^t \sum_{i, \bar{\nu}} \bar{r}_{\bar{\nu}}^{[i]}(t') dt' \right], \quad (10)$$

where $\mathcal{T}$ is the random time of the first particle moving out of $(x, y, c)$, the sum is over all possible directions and all particle index $i \in \{1 \ldots n\}$. Identifying the total rate $\sum_{i, \bar{\nu}} \bar{r}_{\bar{\nu}}^{[i]}(t') dt'$ and the reverse-time transition rate for each particle and in each direction, Eq. (8), we arrived at Eq. (4).

## B. Training and generation algorithms.

Algorithm 1 gives the training algorithm using standard gradient descent techniques, and Algorithm 2 gives the inference algorithm used in this work.

---

**Algorithm 1** DSD training

---

    Given the full transition probabilities $p_t(x', y', c'|x, y, c)$
   **repeat**
      $I_0 \leftarrow$ a sample drawn from the training set
      Draw an index $k$ from $\{1, \dots T\}$ uniformly
      **for** Each discrete intensity unit in $[I_0]_{x,y,c}$ **do**
         Draw $(x', y', c') \sim p_t(x', y', c'|x, y, c)$
         Move the unit from $(x, y, c)$ to $(x', y', c')$
      **end for**
      $I_{t_k} \leftarrow$ the corrupted image
      Compute the reverse transition rate Eq. (4)
      **if** Using $L^1$ rate-matching **then**
         Loss $\leftarrow \sum_{x,y,c,\bar{\nu}} |\bar{r}^{\text{NN}} - \bar{r}|$
      **else if** Using process likelihood **then**
         Loss $\leftarrow -\log L$, defined in Eq. (7)
      **end if**
      Take a gradient step on $\nabla_\theta$Loss
   **until** Converged

---

**Algorithm 2** DSD inference

---

    Given CFL condition number $\varepsilon < 1$ and desired total intensities in the color channels, initiate an image $I_0$ with desired total intensities in the color channels
   **for** Each discrete intensity unit in $[I_0]_{x,y,c}$ **do**
      Draw $(x', y', c') \sim p_1(x', y', c'|x, y, c)$
      Move the unit from $(x, y, c)$ to $(x', y', c')$
   **end for**
   $I_1 \leftarrow$ the fully corrupted image, $t \leftarrow 1$
   **while** $t > 0$ **do**
      Evaluate NN predicted reverse rates $\bar{r}^{\text{NN}}_{\bar{\nu},x,y,c}$
      $\tau \leftarrow \min \left\{ t, \varepsilon \min_{\bar{\nu},x,y,c} \left( \bar{r}^{\text{NN}}_{\bar{\nu},x,y,c} \right)^{-1} \right\}$
      **for** each $(x, y, c)$ **do**
         Sample total moving particles:
         $n_\Sigma \sim \text{Binom} \left( [I_t]_{x,y,c}, \sum_{\bar{\nu}} \bar{r}^{\text{NN}}_{\bar{\nu},x,y,c} \right)$
         Sample a direction $\bar{\nu}$ for each moving particle:
         $n_{\bar{\nu}} \sim \text{Multinomial} \left( n_\Sigma, p_{\bar{\nu}} = \frac{\bar{r}^{\text{NN}}_{\bar{\nu},x,y,c}}{\sum_{\bar{\nu}'} \bar{r}^{\text{NN}}_{\bar{\nu}',x,y,c}} \right)$
         Move $n_{\bar{\nu}}$ intensity units to $(x + \bar{\nu}_x, y + \bar{\nu}_y)$
      **end for**
      Advance time: $t \leftarrow t - \tau$
      $I_t \leftarrow$ the configuration after movements
   **end while**

---

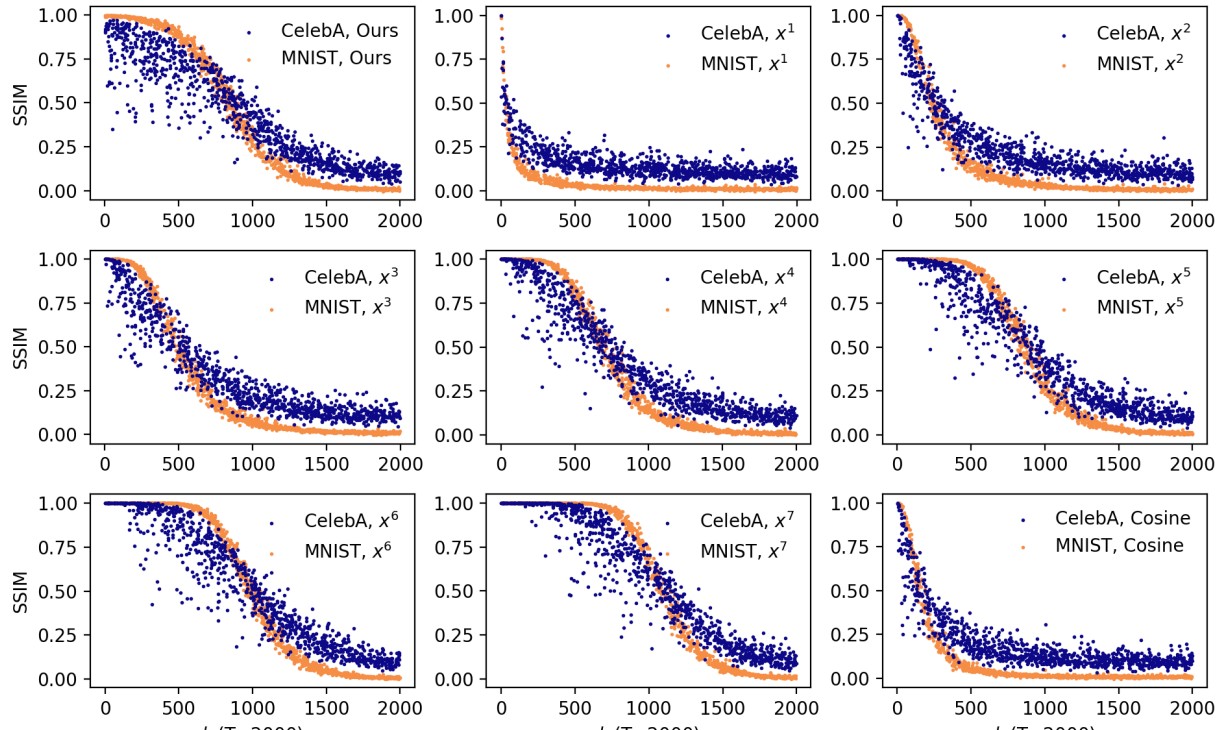

*Figure 6.* Structural Similarity Index Metric between the original and corrupted MNIST and CelebA images (1,000 samples evaluated at uniformly sampled random time $k \in \{1 \ldots 2000\}$) with various noise scheduler. We fixed $r = 120$ for MNIST and $r = 200$ for CelebA in this analysis. For ours, we use Eq. (2) with $\tau_1 = 7.5$ and $\tau_2 = 2.5$. For the polynomials, $t_k = (k/T)^n$, $n = 1 \ldots 7$. For the cosine schedule, we use the heuristic formula given in (Nichol & Dhariwal, 2021). We remark that although the cosine schedule has been shown to be superior in previous studies (Nichol & Dhariwal, 2021; Santos & Lin, 2023), the conclusion is based on the Ornstein–Uhlenbeck process, which is distinct from the spatial diffusion process (1).

## C. Additional MNIST experiments

In our additional MNIST experiments, we explored class-conditional and in-painting generation. These experiments are particularly notable due to their interactions with the mass-preserving property of DSD. For class-conditioning, we introduced class embeddings into our model following the approach described in (Song et al., 2021b). Our model performed well at the task of class retrieval, consistently producing the desired class 8. For our mass-related experiment, we tested our model on its ability to generate all of the classes given different starting masses. Because generative models struggle to extrapolate beyond training data, our model demonstrated poor performance for certain digits on masses that were too high or too low. In response to this, we picked the '1' with the highest mass for our high-mass test, and the '0' with the lowest for our low mass test, as 1 had the lowest mass of any of the numbers, and 0 had the highest. Our model performed very well on this task, consistently producing the target class even with varying mass. See Fig. 3 (d) for results.

In training our model to perform in-painting, we shrunk the size of the transition matrix and held the rest of the image static. We observed high quality generations very quickly, within only 40K training steps. For our mass-related experiment, we tested the model's reaction to increasing mass within the in-painted region and were able to see different number generations from the same starting image (Fig. 9 ).

## D. Hyperparameters for experiments

In our experiments, we thoroughly tested our model on various hyperparameters using the MNIST dataset. The MNIST dataset was chosen as a baseline for hyperparameter testing due to its low computational training cost. We found that our model was very robust with respect to the hyperparameters used, consistently generating quality generations without

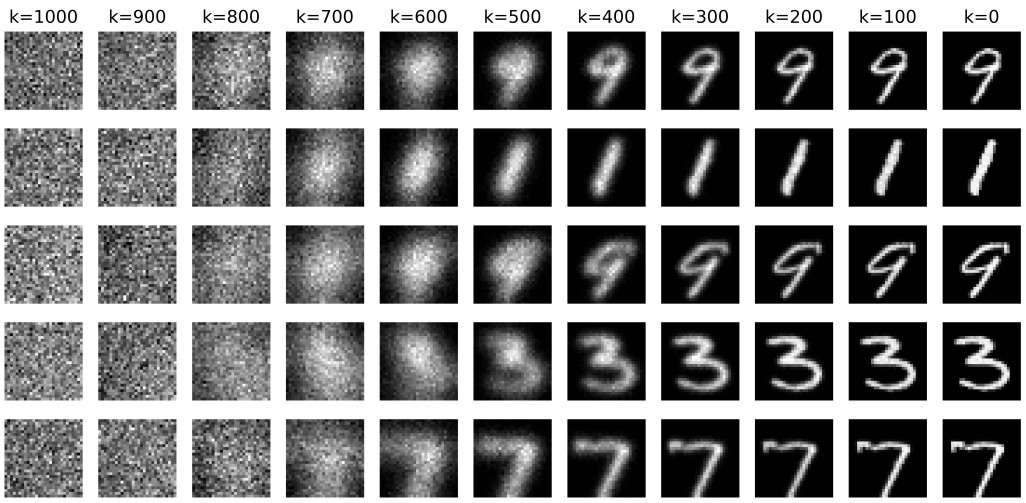

*Figure 7.* Unconditional MNIST generations

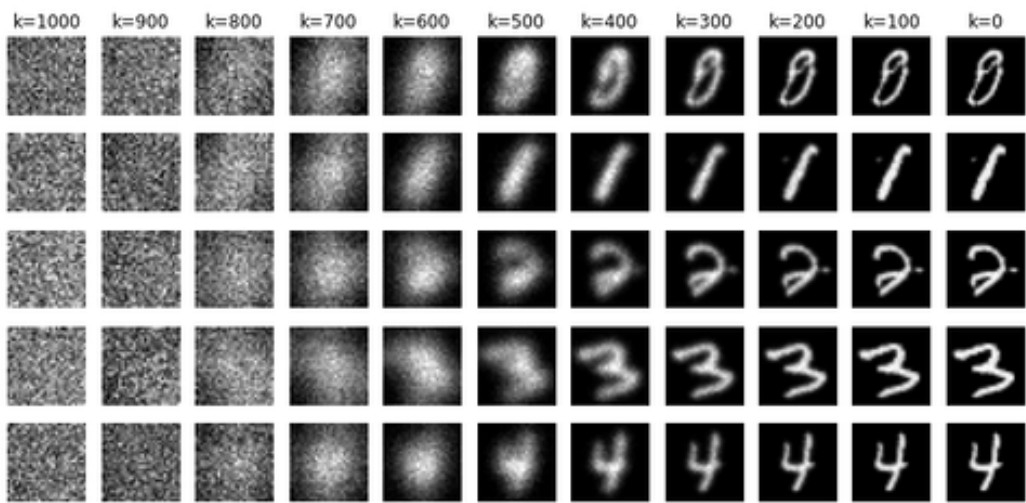

*Figure 8.* Conditional MNIST generations

hyperparameter tuning. Due to limited compute, only limited tests were performed on CelebA, but we hypothesize that our model would perform well with different hyperparameters than the ones used. For the choice of our 'r', we chose a rate that was large enough to allow full degradation, enabling the model to learn to predict starting from full noise. See 1 for our hyperparameters used.

## E. Detailed description of X-ray scans of subsurface rocks

- **Berea Sandstone**: This sandstone sample from (Neumann et al., 2021) provides a high-resolution image of the rock microstructures obtained through X-ray microtomography (X-ray $\mu$CT). In this process, the rock sample is rotated while being scanned by an X-ray beam, capturing a series of 2D radiographs at different angles. These projections are then computationally reconstructed into a 3D volume, where each voxel represents the X-ray attenuation of the material at that location. The X-ray microtomography scans were performed using a SkyScan 1272 system, operating at 50 kV and 200 $\mu$A, with a CCD detector capturing projections at a resolution of 2.25 $\mu$m per voxel. The resulting dataset consists of grayscale images with a voxel size of 2.25 $\mu$m, where variations in intensity distinguish between the solid matrix and the pore space. The solid matrix primarily consists of tightly packed mineral grains—mostly

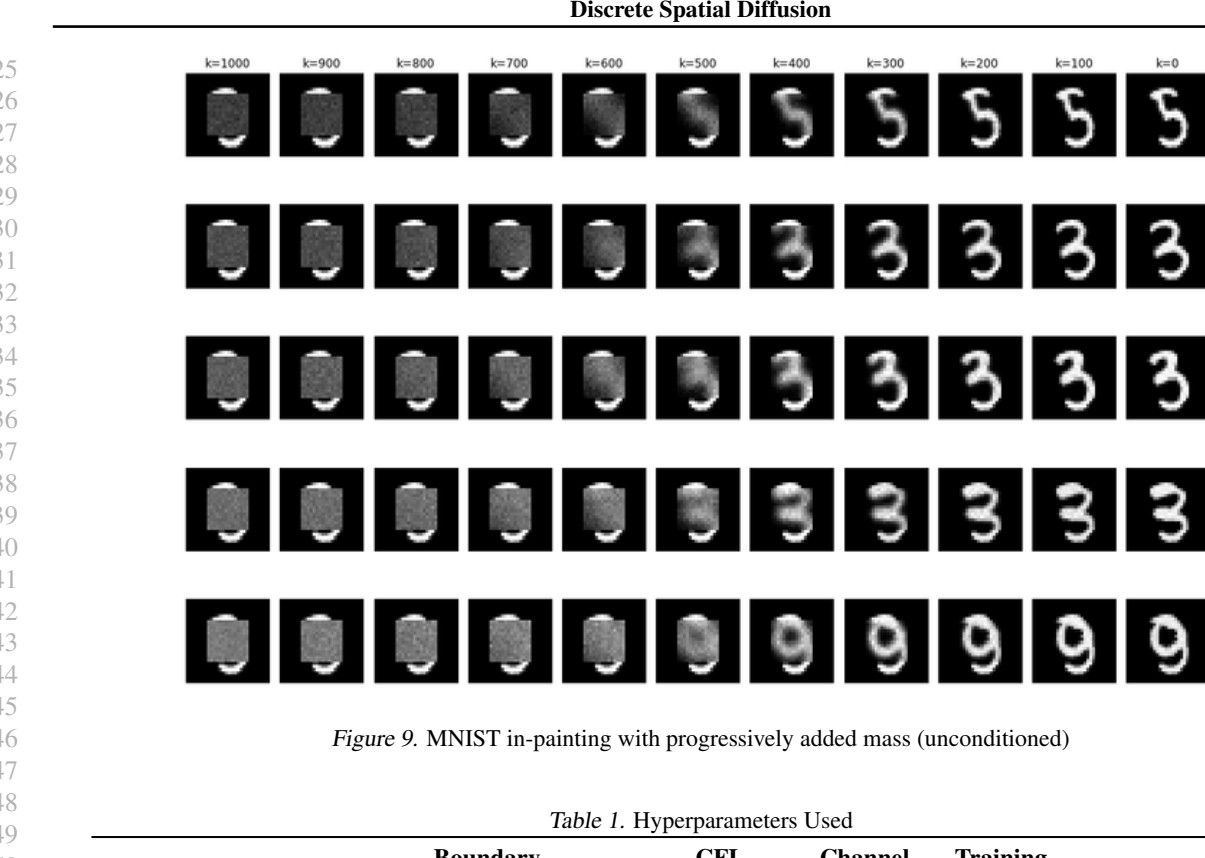

*Figure 9.* MNIST in-painting with progressively added mass (unconditioned)

*Table 1.* Hyperparameters Used

| Dataset | $r$ | Schedule | Boundary Condition | Loss | CFL Tolerance | Channel Multiplier | Training Iterations | Notes |
|---------|-----|----------|--------------------|------|---------------|--------------------|--------------------|-------|
| MNIST | 120 | Ours | Periodic | Eq. 5 | 0.15 | (2,2,2) | 100K | unconditional |
| MNIST | 120 | Ours | Periodic | Eq. 6 | 0.15 | (2,2,2) | 90K | unconditional |
| MNIST | 120 | Ours | No-flux | Eq. 6 | 0.15 | (2,2,2) | 80K | unconditional |
| MNIST | 120 | Ours | No-flux | Eq. 6 | 0.11 | (2,2,2) | 70K | class-conditioned |
| MNIST | 85 | Ours | No-flux | Eq. 6 | 0.07 | (2,2,2) | 40K | inpainting (14x14) |
| CelebA | 200 | Ours | No-flux | Eq. 5 | 0.1 | (1,2,2,2) | 700k | |
| Electrodes | 200 | $x^5$ | Periodic | Eq. 6 | 0.01 | (1,2,2,2) | 180k | |
| Rocks | 250 | $x^4$ | Periodic | Eq. 6 | 0.1/0.2/0.05 | (1,2,2,2) | 50k | tolerance avoids over-lapping mass |

quartz—while the pores are voids that can be occupied by fluids such as water or hydrocarbons. After preprocessing steps like contrast enhancement, noise reduction, and segmentation, the final dataset represents the pore network. The Berea sample has a measured porosity of 18.96% and permeability of 121 mD. This dataset is particularly useful for computational modeling, as it enables direct comparison between numerical simulations and experimentally measured permeability, providing a rich testbed for learning-based methods that seek to map complex microstructural information to macroscopic transport properties. This sedimentary rock is a well-characterized geological benchmark, widely used in fluid flow studies due to its homogeneous grain structure and consistent permeability properties, making it a good first benchmark for our study.

- **Savonnières Carbonate**: This carbonate sample, described in (Bultreys et al., 2016), is a layered, oolithic grainstone with a wide porosity and a permeability varying from 115 to over 2000 mD, depending on local heterogeneities. The rock is characterized by a highly multimodal and interconnected pore structure, with distinct macropores and microporosity. X-ray microtomography (X-ray $\mu$CT) was used to image the sample at a resolution of 3.8 $\mu$m voxel size, revealing intricate pore geometries. The sample was scanned at the Ghent University Centre for X-ray Tomography (UGCT) using their HECTOR scanner, developed in collaboration with XRE, Belgium. The macropores include both intergranular voids and hollow ooids, while the microporosity is found within ooid shells and intergranular spaces. Micropores in the sample often serve as the primary pathways connecting poorly connected macropores, creating a

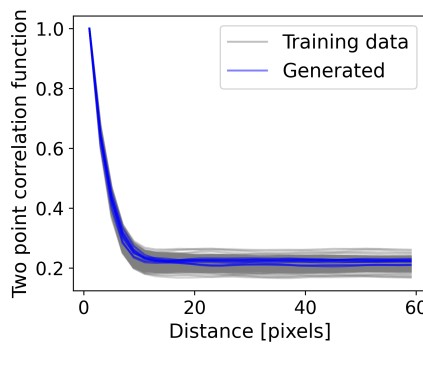
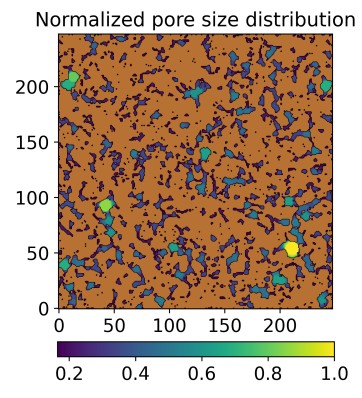
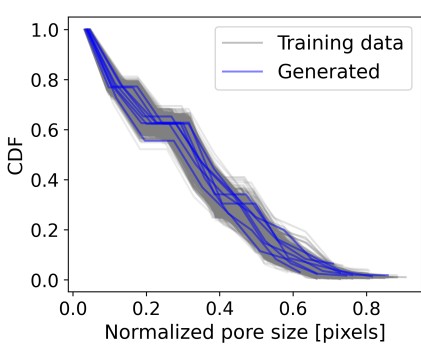

*Figure 10.* Quantitative comparison between training and generated rock samples. **(Left)** Two-point correlation function, showing excellent agreement of spatial features between training data (gray) and 10 randomly generated samples (blue). **(Middle)** Normalized pore size distribution schematic, with colors indicating relative pore sizes. **(Right)** Cumulative distribution function (CDF) of the normalized pore sizes, comparing the statistical distribution of training and generated samples.

complex hierarchical network. After preprocessing steps, including noise reduction, anisotropic diffusion filtering, and watershed segmentation, a multiscale pore network model was extracted. This dataset is particularly compelling due to its extreme heterogeneity, with pore sizes spanning orders of magnitude, and its ability to represent coupled serial and parallel flow pathways. Savonnières serves as a test case for studying the impact of complex samples in our workflow.

- **Massangis Limestone**: This oolitic limestone sample from (Boone, 2014) is a highly heterogeneous carbonate rock with a complex, multimodal pore structure resulting from diagenetic alterations, including dolomitization and dedolomitization. The rock contains a mix of intergranular and moldic macroporosity, along with microporosity concentrated in ooid rims and partially dissolved dolomite regions. Its porosity ranges from 9.5% to 13.8%, depending on local variations, and its permeability is highly anisotropic due to the interplay between connected macropores and poorly accessible microporosity. X-ray microtomography (X-ray $\mu$CT) was used to image the sample at a voxel resolution of 4.54 $\mu$m, capturing the intricate connectivity of macro- and micropores. The sample was scanned at the Ghent University Centre for X-ray Tomography (UGCT) using a FeinFocus FXE160.51 transmission tube, in collaboration with Paul Scherrer Institute (PSI), Switzerland. Differential imaging was applied to enhance the detection of fluid-filled microporosity, revealing the rock's internal heterogeneities. Unlike more uniform carbonate samples, Massangis exhibits significant spatial variations in pore connectivity, leading to zones of high permeability interspersed with isolated pore networks. This dataset serves as another challenging benchmark for modeling porous media microstructure.

### E.1. Effective metrics

In porous media analysis, characterizing the spatial arrangement and size distribution of pores is crucial for understanding transport properties, mechanical behavior, and overall structure-function relationships. To quantify these characteristics, we compute the spatial correlation function and pore size distribution (PSD) using PoreSpy (Gostick et al., 2019), a Python-based toolkit for quantitative analysis of porous media images. The **Pore Size Distribution (PSD)** characterizes the variation of pore sizes within a porous material, providing insights into connectivity, permeability, and flow dynamics. The most common method to determine PSD computationally is the local thickness approach. Given a binary image $I(x, y)$, where pore space is represented as 1 and solid space as 0, the pore size function $f(r)$ is defined as the probability density function (PDF) of the largest sphere that can be inscribed at any point within the pore space. The PSD provides a statistical summary of pore connectivity and transport properties. Small pores dominate permeability, while large pores govern bulk flow. Both of these metrics for the training and generated samples are shown in Figure 10.

## F. X-ray scans of NMC cathodes

### F.1. Dataset description

This dataset provides high-resolution 3D images of a Li-ion battery cathode composed of active material (nickel-manganese-cobalt oxide, NMC), carbon black, and a polymer binder (Usseglio-Viretta et al., 2018). The cathode sample was imaged via X-ray microtomography (X-ray μCT) and nano-tomography (X-ray nano-CT) to capture both the overall electrode architecture and fine-scale features of the carbon/binder domain (CBD). For micro-CT, a Zeiss Xradia Versa 520 system was operated at 80 kV and 88 μA, acquiring projections at an effective isotropic voxel size of approximately 398 nm over a field of view of about 400 μm. The nano-CT scans were performed using a Zeiss Xradia Ultra 810 system with a chromium target (35 kV, 25 mA), yielding isotropic voxel sizes on the order of 126 nm across a field of view of approximately 64 μm. In both cases, the 2D radiographs were reconstructed into 3D grayscale volumes using a filtered back-projection algorithm, capturing the X-ray attenuation due to the dense NMC particles and the less attenuating pore/CBD regions.

These tomographic datasets reveal the hierarchical microstructure of the electrode, from tens-of-micrometers NMC active particles to nanometer-scale pores within the percolated carbon network. After preprocessing—such as non-local mean filtering, contrast enhancement, and slice-by-slice local thresholding—segmentation identifies three main phases: (1) the NMC active material, (2) the CBD, and (3) the pore space. Measured porosity values for these cathodes can exceed 30%, while the typical volume fraction of active material is on the order of 40%. The overall areal loading of the active material is around 29.78 mg·cm$^{-2}$, corresponding to about 33 mAh·cm$^{-2}$ in specific capacity. These 3D reconstructions enable computational modeling of transport properties (e.g., tortuosity factor) and electrochemical performance, facilitating direct comparisons with experimentally measured parameters. Because of the electrode's well-defined spherical NMC particles and percolating carbon network, this dataset serves as a robust benchmark for multi-scale modeling and data-driven methods that aim to link microstructural features to macroscopic cell behavior.

### F.2. Effective metrics

The analysis of NMC cathode tomography and the generated images was conducted using three metrics: interface length, triple-phase boundary, and relative diffusivity. These metrics are essential for quantifying the morphological and transport characteristics that influence the electrode's electrochemical performance. Below we describe these metrics in detail.

**Interface length** refers to the total length of boundaries where two distinct phases, such as active material and pore or electrolyte, intersect. A higher interface length indicates more active sites for electrochemical reactions and enhances ion transport pathways, thereby improving the electrode's overall performance. This metric is calculated by identifying and summing the perimeters of all phase boundaries in the segmented image.

**Triple-Phase Boundary** denotes the regions where three different phases—typically solid active material, electrolyte, and a conductive phase or pore space—converge in the microstructure. TPBs are crucial for facilitating efficient electrochemical reactions, as they provide optimal sites where all necessary phases interact. The total TPB length is determined by locating points or lines where three phases meet and summing their lengths within the image.

**Relative Diffusivity** quantifies the reduction in ion transport within the porous cathode structure relative to an unobstructed medium. It is defined as the ratio of the effective diffusivity, $D_{\text{eff}}$, through the porous medium to the intrinsic diffusivity, $D_0$, of the conductive phase: $D_{\text{rel}} = D_{\text{eff}}/D_0$. This reduction is primarily attributed to the geometric complexities of the microstructure, encapsulated by the tortuosity factor, $\tau$, in fact $D_{\text{rel}} = D_{\text{eff}}/D_0 = V_f/\tau$, where $V_f$ is the volume fraction of the phase under analysis.

We computed these metrics using the Python library TauFactor (Kench et al., 2023), and the comparisons between the real and generated images based on these metrics are illustrated in Fig. 12, while a collection of the training data and generated images is in Fig.11.

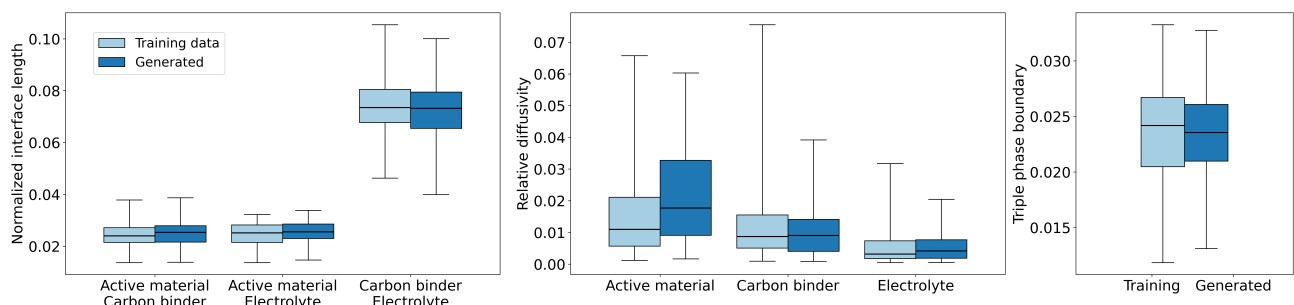

*Figure 11.* Microstructural characterization metrics for 80 training samples and 80 generated samples. The boxes show the 25th-50th-75th percentile, the whiskers the minimum and maximum values. Metrics computed using TauFactor (Kench et al., 2023).

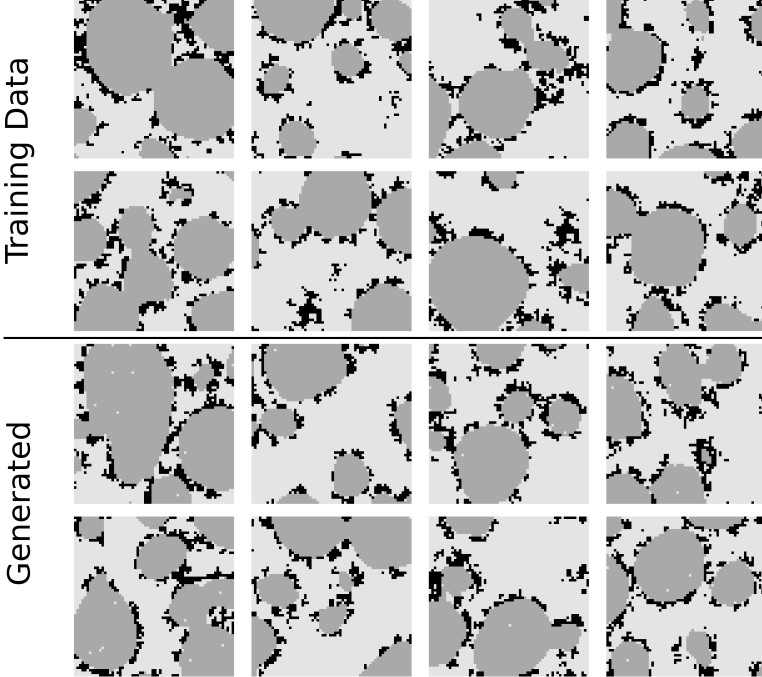

*Figure 12.* (Top) Eight randomly picked samples from the NMC cathodes dataset. (Bottom) Random unconditional realizations of our model.

