# OpenReview forum: "Discrete Spatial Diffusion: Intensity-Preserving Diffusion Modeling"
_ICML.cc/2025/Conference — Submitted to ICML 2025_

### Official Review · Reviewer_V6gu · 2025-02-25

**Overall Recommendation:** 4

**Summary:**

The authors proposed a discrete diffusion model that respects mass conservation. They define the forward process as a random walk of particles and train an NCSN to reorganize these particles during the reverse process. The total mass of the particles is preserved since these operations do not create or destroy particles. While the proposed DSD model is designed for applications in material microstructure, the authors also demonstrate its effectiveness on image datasets.

**Claims And Evidence:**

The claims are clear.

**Essential References Not Discussed:**

Not that I am aware of.

**Experimental Designs Or Analyses:**

I checked the experimental results and they are clear to me. Perhaps the authors can consider comparing the FID between DSD and other Gaussian-noise diffusion models. I understand that generating high-quality images is not the main goal but this helps understand the behavior of DSD.

**Methods And Evaluation Criteria:**

The method and evaluation make sense to me.

**Other Comments Or Suggestions:**

Fig. 1(c) should be fixed. See my other comments above.

**Other Strengths And Weaknesses:**

Strengths: The proposed method is novel and valuable for scientific use in specific domains. The mass-preserving property is well-demonstrated by the experiments.

Weaknesses: While the paper is mostly easy to follow, at places I do feel some paragraphs are lengthy and overwhelming. For example, breaking down the relationships between DSD, FPE, and heat equation can help the reader better understand the model.

**Questions For Authors:**

1. In Fig. 4, the differences between the generated representations are not obvious. Does the total mass of these datasets have small variances?
2. How will existing discrete diffusion models perform on the tasks if the mass is provided as a condition? Adding such comparisons can better demonstrate the difficulty of the tasks.

**Relation To Broader Scientific Literature:**

The proposed method is valuable for specific physics applications.

**Theoretical Claims:**

The derivation of DSD hinges on (Santos et al., 2023). The authors did not provide other theoretical results.

---

> ### Author Rebuttal · Authors · 2025-04-01
>
> We thank the reviewer for their careful assessment of our results.
>
> **4.1** [*Theoretical results*]  We agree that the core derivation of DSD builds on the framework of Santos et al. (2023). That said, we submit that the paper includes theoretical contributions, albeit specific to DSD. These include the derivation of first-reaction rates (Section 3.3 and Appendix A), as well as the Green’s function (propagator) for the DSD process under periodic boundary conditions (Appendix C). These mathematical results are critical for enabling efficient and exact computation within the DSD framework.
>
>
> **4.2** [*FID comparisons*] We agree that this helps comparability. We have found the following preliminary FIDs:
> | Dataset             | FID  | #   | Comments                           |
> |---------------------|------|-----|------------------------------------|
> | MNIST               | 5.73 | 8K  |                                    |
> | CIFAR-10            | 53.7 | 14K |                                    |
> | CIFAR-10            | 40.9 | 14K | Mean-preserving Gaussian filter[*] |
> | CelebA-64           | 60.1 | 8k  |                                    |
> | CelebA-64           | 41.9 | 8k  | Mean-preserving Gaussian filter[*] |
> | Rock microstructures| 0.9  | 300 |                                    |
> | Batteries           | 1.3  | 300 |                                    |
> [*] While FID is not the primary goal of DSD, its pixel-by-pixel process leads to less smooth outputs and higher FIDs; applying a mean-preserving Gaussian filter improves FID by aligning better with image-based metrics.
>
> We have also performed additional quantitative metrics for our models, as well as comparisons to existing techniques, please see points **1.2** and **2.2**.
>
> **4.3** [*Lengthy paragraphs*]  We appreciate the feedback. If accepted we will clearly expand the explanation of how DSD links to Fokker–Planck and heat equations using some of the additional space, and we will reorganize lengthy paragraphs for better clarity. We are also willing to include other detailed derivations and/or explanations that the reviewer requests in the appendices.
>
> **4.4** [*Fixing Fig. 1(c)*]  We are unsure about the specific concern the reviewer is raising regarding Fig. 1c, and would appreciate clarification to better address this comment.
>
> **4.5** [*Differences in generations of Fig. 4*] Thank you for this helpful question. In response, we have added a new visualization that shows the distribution of the training data for the rock microstructure dataset. Specifically, we plot the sampling distribution of $\mu$, along with $\mu \pm 2\sigma$ and $\mu \pm 3\sigma$ to better illustrate the spread of the generated samples. This updated figure more highlights our model's ability to generate samples across the full range of the dataset’s variability, maintaining consistent structure even at the distributional tails. The updated figure can be found [here](https://rb.gy/mdmzam).
>
> We also contrast our approach with [recent work](https://rb.gy/8x23e3), led by [Professor Martin Blunt](https://rb.gy/ru98qg). Their model lacks explicit control over porosity, and so it produces out-of-distribution generated samples, with longer tails in porosity distribution (see their Figure 3E). Our DSD offers a clear advantage when precise field-measurement tolerances of multiple significant digits have to be met. We will revise the manuscript to briefly include this very recent work and clarify how our model addresses these limitations.
>
>
> **4.6** [*Comparison with mass conditioning approaches*]  In order to respond to this point, we have evaluated a baseline discrete diffusion model conditioned on the target total intensity (or mass). The model performs reasonably well near the mean of the dataset, where the target intensity falls clearly within the training distribution. However, it tends to reach this point by producing small negative values in the final image—despite being trained on images with pixel values in the [0-1] range. When we threshold these negative values to zero and discretize the image to the [0, 255] range, the intensity error increases. This degradation becomes especially pronounced in the tails of the intensity distribution. In these regions, the Gaussian diffusion model often fails to honor the conditioning and produces non-compliant samples. In contrast, our method (DSD) guarantees intensity compliance across the full range, including the tails of the distribution. This difference is demonstrated in this [Figure](https://rb.gy/dyuymu). Please also see response **1.4**  where we compared to another approach.
>
> **4.7** [Summary] We have addressed the reviewer’s clarification to the best of our ability, added further quantitative comparisons, and better explanation of the significance for Fig. 4. We believe that addressing these comments substantially improves the manuscript.

---

> > ### Comment · Reviewer_V6gu · 2025-04-07
> >
> > I thank the authors for their detailed responses. The right-most grid in Fig. 1(c) is not plotted properly, but my other concerns have been addressed. I have updated my score to 4 accordingly.

---

### Official Review · Reviewer_7xJk · 2025-03-12

**Overall Recommendation:** 2

**Summary:**

This paper introduces Discrete Spatial Diffusion (DSD), a new diffusion-based generative modeling approach that operates on discrete intensity units and enforces strict global conservation of these intensities. Traditional image-based diffusion models typically treat pixel intensities as continuous quantities, and independently diffuse intensities at each pixel. In contrast, DSD places each intensity “unit” on a spatial lattice and performs a continuous-time random walk of these units, which ensures that the total intensity per color channel does not change throughout both the forward and the reverse processes.

**Claims And Evidence:**

The paper claims exact preservation of total intensity. The construction as a jump process enforces the claim since the entire chain simply redistributes discrete “particles.”
The paper claims its ability to generate coherent images, which can be seen by example images on MNIST, CelebA, and microstructure data. Some samples look qualitatively plausible, which partially supports this claim.
The authors present domain-specific examples in section 4 (lithium-ion electrode phases, rock porosity) where they replicate morphological features. This is evidence for the method’s applicability, though the paper’s evaluations remain primarily qualitative or rely on morphological measures.
The experimental comparisons are relatively narrow, especially for standard image benchmarks, there is no evidence how well the approach outperforms simpler baselines on domain tasks.

**Essential References Not Discussed:**

the paper cites enough references

**Experimental Designs Or Analyses:**

domain-specific datasets (rock microstructures, lithium-ion electrode scans). For MNIST and CelebA, they demonstrate unconditional generation, class conditioning, and inpainting, but they do not report common generative metrics like FID or IS. Instead, they rely on visual inspection and qualitative comparisons for these datasets.

For microstructure data, they measure morphological properties relevant to the domain (for example, porosity, phase-volume fractions). Although they do not perform much large-scale physical simulations, they argue these morphological analyses are indicative of fidelity in scientific contexts.

 Potential Issues:
The evaluation on standard benchmarks is more limited than typical generative model papers (less emphasis on FID).
Scalability to larger images and verifying performance with thorough domain simulations remain open challenges. While the paper acknowledges these limitations, it does not fully address them.

**Methods And Evaluation Criteria:**

The proposed method, DSD, is formulated as continuous time, discrete-state Markov jumps on a spatial lattice. It uses network architecture adapted from NCSN++ to predict reverse transition rates. For evaluation, the authors give primarily qualitative (sample visual quality) plus some morphological or porosity-based statistics in scientific domains, as well as they check the inpainting capability if partial images yield coherent reconstructions.
The paper does not provide in-depth baseline comparisons and widely used generative metrics (FID). Thus, it is unclear how DSD’s sample quality or training efficiency compares to continuous diffusion methods.

**Other Comments Or Suggestions:**

A table comparing training time, sampling speed, and sample quality (FID-like metric) versus standard diffusion would clarify the practical trade-offs.

**Other Strengths And Weaknesses:**

Strengths:
This paper demonstrates general utility across standard datasets (MNIST, CelebA) and domain-specific tasks.
This paper gives a clear derivation of the proposed method, DSD.

Weakness:
The proposed method may encounter computational complexity for large, high-resolution images. Since each intensity unit is an explicit “particle,” the forward corruption and backward sampling can become expensive for typical 8-bit or 16-bit images with large spatial resolutions. Although the paper acknowledges this constraint, it offers no clear strategy for extending the approach to high-res images.
In experiments, the authors only give limited benchmark comparisons. For example, performance metrics like FID are less explored here in CelebA dataset. The paper focuses on domain-specific usage, but it reduces direct comparability.
This paper does not attempt to claim better coverage or better generative fidelity than standard continuous diffusion and no exact SOTA claims or comparison on typical image tasks, which is less convincing.

**Questions For Authors:**

How to handle large-scale images with higher bit-depth? Could partial binning or approximate methods still preserve approximate mass constraints?
Could you provide standard generative metrics (FID) for comparability? Even though your focus is on constraints, I hope to see standard benchmarks.

**Relation To Broader Scientific Literature:**

For Generative Diffusion Models, the paper places itself as discrete diffusion approaches (Hoogeboom et al., Austin et al., Campbell et al., etc.) and with heat-based or spatially correlated noise approaches such as Inverse Heat Dissipation or Blurring Diffusion.
For scientific microstructure modeling, they cite domain-specific references dealing with microstructures. They state that the proposed DSD can preserve global constraints exactly while the references cannot.

**Theoretical Claims:**

There are no theoretical claims in this paper.

---

> ### Author Rebuttal · Authors · 2025-04-01
>
> We thank the reviewer for their constructive comments; we feel that the reviewer has understood the purpose of the work and offers concrete avenues for improvement, all of which we are able to address in our revision.
>
>
> **3.1** [*computational complexity*] First, let us remark on the computational requirements, per iteration, involved in DSD. Noising the images involves sampling using a random walk propagator that is precomputed–this is cheap enough to be amortized by dataloader workers on CPU, so training DSD is not any more expensive than training NCSN++. At test time with our current non-parallel code, about half the time is spent moving the particles around and half the time at network evaluation. The overall time for training is similar to other diffusion models. Sampling time depends on the noise schedule and CFL tolerance parameter. Good samples can presently be accomplished in <4 minutes per celebA sample per GPU without any special software engineering and A5000 hardware (this includes batching of images). In terms of cost breakdown, on celebA (the most challenging for DSD in this work), roughly equal amount of time is required for moving particles as performing NCSN++ evaluation; the extra computational cost for DSD is approximately a factor of 2, and in our inference code this (CPU-based) cost is presently hidden by running multiple inferences in parallel processes. We will revise the manuscript to describe this information.
>
>
> **3.2** [*high resolution*] Regarding scaling, our method scales effectively to larger resolutions, and as a demonstration, we have trained it on [1000×1000 Leopard sandstone images](https://rb.gy/3wsudt): realizations are shown in the attached repository as well as included in the revision. Leopard sandstones feature larger critical percolation diameters and more dispersed clay regions. Our model captures these morphological nuances, demonstrating both fine-scale and macro-scale structural details.
>
>
> **3.3** [*scalability*] The reviewer’s point that DSD may be difficult to scale to images which are both high resolution and high bit depth (high-bit-depth) images. As mentioned (**3.1**), sampling noised images is not expensive–and so this is a concern only for inference throughput. We do address 24-bit CelebA images and separately 1000x1000 rock images (**3.2**). We feel that DSD is applicable to a variety of datasets, and future innovations may make it possible to scale the approach further. It may only be a question of implementation that is solvable with a GPU kernel for particle motion sampling.
>
>
> **3.4** [*quantifying performance*] Regarding quantifying performance, we have primarily focused on domain-specific measures (see also **2.2**) because these binary microstructure images differ from typical computer vision benchmarks. Nonetheless the reviewer’s point about comparability is well-taken, and so we now include FID metrics for MNIST and CelebA. It is important to take these in context, as although the performance of these models with respect to FID is not state-of-the-art, our intent is not to replace the state of the art, but to address the niche where conservation of intensity/particles is necessary. To that end, we also offer to demonstrate the ability for a conditional standard gaussian diffusion model to produce given overall intensities on MNIST (**1.4**). The conditioning works well within the typical range of intensities for the dataset, but falters very significantly as one approaches the tails of intensity. Even still, the conditioned model sometimes produces *negative* intensities in order to meet its objective for total intensity; this failure mode is not possible with DSD. All of this will also be described in our revision.
>
>
> **3.5** [*standard benchmarks*] To further address the reviewer’s comment that they would like to see more standard benchmarks: We would like to argue that the non-standard tasks we introduce (intensity constrained generation and intensity constrained inpainting) provide value to the field as well; while one goal for ML research is to advance the state of the art for identified important tasks, another important goal is to advance the scope of (well-motivated) tasks for which ML techniques are available.
>
>
> **3.6** [*summary*] Through these revisions and clarifications, we are confident that our revised version will address the reviewer’s concerns about quantitative comparability, computational costs, and scalability. While our models do not advance state-of-the art FID scores, DSD is reasonably affordable, solves a well-motivated novel scientific task (intensity constrained generative modeling) far better than simple extensions of existing work (input-based conditioned models, and c.f. **1.4** regarding another approach), and produces very high quality domain metrics (**2.2**, **4.2**, **4.6**)

---

### Official Review · Reviewer_99qB · 2025-03-14

**Overall Recommendation:** 3

**Summary:**

This paper presents Discrete Spatial Diffusion (DSD), a framework that ensures intensity preservation in diffusion models by using a continuous-time, discrete-state jump stochastic process. Unlike standard diffusion models that operate in continuous intensity spaces, DSD naturally incorporates stochasticity while maintaining conservation laws, making it well-suited for scientific applications.

**Claims And Evidence:**

- Problematic Claim:
1. The model’s capability to control phase volume fractions in scientific applications (e.g., lithium-ion electrodes, porous rock microstructures) is supported by qualitative results.
2. The authors demonstrate that such a model is powerful enough for conventional image synthesis tasks.

- Reasons:
1. While qualitative examples are provided, quantitative evaluations (e.g., comparison with existing vanilla diffusion models w.r.t. FID score) are limited.
2. The paper does not clearly demonstrate whether the model generalizes well on complex datasets as the unconditional generation outputs of CelebA are not good.

**Essential References Not Discussed:**

The paper has referenced all the important related works.

**Experimental Designs Or Analyses:**

The experimental design is reasonable and aligns well with the paper’s objectives. The evaluation effectively demonstrates the model’s ability to preserve intensity in discrete-state diffusion and explores its applications in scientific contexts (Figures 4 and 5) and image synthesis tasks (Figure 3).

MNIST, as a structured image dataset, is well-suited for assessing the model’s ability to preserve spatial features. Similarly, rock subsurface and lithium-ion electrodes are materials with complex microstructures, where intensity preservation is essential for accurately modeling pore structures and phase distributions. These dataset choices effectively showcase the applicability of the proposed method across both structured image data and scientific materials.

**Methods And Evaluation Criteria:**

The method effectively preserves intensity and structure in discrete-state diffusion models, making it suitable for simple image synthesis and scientific applications such as porous materials and lithium-ion battery electrodes.

However, the evaluation primarily relies on qualitative results (e.g., visualizations in Figures 3 and 4). To enhance the analysis, the authors should incorporate quantitative metrics (e.g., FID, MMD, or SSIM) to compare generated microstructures with real samples. Although Figure 6 presents a time schedule design inspired by SSIM, the paper lacks a specific quantitative comparison with other benchmarks.

**Other Comments Or Suggestions:**

I have no additional comments or suggestions.

**Other Strengths And Weaknesses:**

This paper presents a highly novel approach with a clear structure and well-organized presentation, making it easy to follow. It provides a thorough review and comparison of prior work while offering a comprehensive summary of existing methods. Notably, it is the first to introduce intensity preservation in diffusion models using a continuous-time, discrete-state jump stochastic process, with significant implications for medical imaging, astronomical data synthesis, super-resolution reconstruction, and advancements in film effects and rendering.

**Questions For Authors:**

See above.

**Relation To Broader Scientific Literature:**

This paper builds on prior work in discrete-state diffusion models, intensity-preserving generative processes, and scientific applications:
1. Discrete-State Diffusion Models

Extends traditional diffusion models (Ho et al., 2020; Song et al., 2021) by introducing a spatially correlated discrete-state process, addressing limitations of prior Markov chain-based models (Hoogeboom et al., 2021; Austin et al., 2021).

2. Intensity-Preserving Generative Processes

Unlike GAN- and VAE-based approaches (Duquesnoy et al., 2023), this method strictly preserves intensity without requiring post-hoc projections or approximations (Chung et al., 2022; Finzi et al., 2023).

3. Scientific Applications

Advances generative modeling for rock subsurface (Blunt et al., 2013) and lithium-ion electrodes (Usseglio-Viretta et al., 2018), enabling precise control over porosity and phase distributions.

**Theoretical Claims:**

1. The paper claims that its discrete-state diffusion process strictly preserves total intensity throughout the forward and reverse processes via particle transition at rate $r$. This theoretical formulation appears sound. However, in Eq. (5) or (7), when measuring the difference between the predicted and true rates ($\bar{r}^{NN}$ and $\bar{r}$), why does the paper use $\bar{r}^{NN} - \bar{r}\log\bar{r}^{NN}$ instead of the more straightforward $\bar{r}^{NN} - \bar{r}$? Could the authors clarify the motivation behind this formulation and its impact on the optimization process?

2. The use of SSIM for designing time schedules is an interesting idea, but the paper does not formally prove why SSIM is an appropriate guiding metric for diffusion noise scheduling. From Figure 6, the polynomials schedule ($x^5$, or $x^6$) also looks good. A deeper mathematical justification, beyond empirical observation, would help support this claim.

---

> ### Author Rebuttal · Authors · 2025-04-01
>
> We thank the reviewer for providing a clear review that shows understanding of the work, constructive criticism for improvement, and good questions for clarification.
>
>
> **2.1** [*incorporate quantitative metrics such as FID*] We emphasize that DSD primary advantage is for scientific applications, whereas the FID metric implicitly focuses on human-centric datasets (the Frechet distance is computed as the latent embedding of the Inception V3 model, which was pre-trained on the ImageNet dataset). That said, we have performed the computation of FID as requested, achieving results in rebuttal section **4.2**. We do not have any rational or theoretical foundation that the FID metric would be meaningful when applied to scientific datasets, however, for completeness sake we will now provide FID to the scientific images as rendered in RGB in the paper. In addition to FID, please see also **1.4** regarding tests comparing Gaussian models to ours with regard to total intensity generation targets.
>
>
> **2.2** [*scientific applications… supported by qualitative results*] The reviewer contends that our quantitative results are limited. In addition to the FID metrics included in revision **2.1**, we do already provide several quantitative measurements highly relevant for the scientific data. For one, porosity (phase volume fraction) is a key variable because permeability of porous media typically scales by the inverse cube of the porosity, and so quantifying our ability to match the total intensity is a relevant metric for scientific applications. Additionally, we present several quantitative scientific evaluations in appendices (due to space constraints).  These come from earth science, materials science, and energy storage literature, and include two-point correlation functions and pore size distributions (Appendix F.1), as well as interface length, triple-phase boundary, and relative diffusivity (Appendix G.2). For context on the relevance of these metrics, please see the paper [Gayon-Lombardo et al 2020](https://rb.gy/f90gtc). Thus we argue that the quantitative evaluation of our models has not been quite limited as the reviewer has claimed.  Perhaps this was not emphasized enough in the main text, and we will revise it to emphasize the domain science metrics.
>
>
> **2.3** [*motivation for loss function*] Your question about the loss functions is a good one. We propose to add the following explanation in revision: `We remark that Eq. (5) is a heuristic approach to match the predicted rate and the ground-truth one, which is a popular approach for building loss functions in machine learning (similar approaches include flow-matching and score-matching); the motivation for using this loss is simplicity and analogy to existing work. Eq. (7) is a more principled  statistical approach, derived in Santos 2023 by designing a maximum likelihood loss using the analytical reverse transition rates. While less intuitive, it is easy to verify by taking ordinary derivatives that the loss is an absolute minimum when the predicted and true reverse rates are equal, the process of which also reveals that this Eq. (7) is essentially the integral of the mean absolute percentage error. We have tested both approaches, and did not observe a significant difference, which demonstrates the robustness of the DSD framework.` We hope this clarifies the nature of the logarithmic loss function.
>
>
> **2.4** [*SSIM-based scheduler*] Indeed, our SSIM-based approach is a heuristic way to construct the time scheduler, which we have mentioned in the manuscript (lines 216-217). We share the sentiment that it would be nice to have a theoretically sound way to construct the scheduler for DSD, but we are not aware of an existing approach which fulfills this criteria, and it is not an innovation that we are able to offer at this time. In revision we will remark on the pursuit of a deeper mathematical support for the time sampling schedule as a good candidate for future work. Ultimately, the design of schedules (noise schedule, time schedule, learning rate schedule) is a grand challenge for which there is no existing framework with a solid and complete mathematical justification (as far as we know).
>
>
> **2.5** [*summary*] We appreciate the reviewers comments, and will be able to address them in revision; we will improve the quantitative comparison using FID score, we will clarify the loss function, and remark further on the heuristic nature of our scheduling and prospects for future work. We also gently argue that our present results are not only qualitative, as we have applied several significant domain-based quantitative metrics. Although we concede that the [CelebA faces](https://rb.gy/p37qwg) fall short of the state of the art (and now quantifyably so), we believe that by addressing the rest of the reviewer’s concerns we have substantially improved the manuscript.

---

> > ### Comment · Reviewer_99qB · 2025-04-03
> >
> > Thank you to the authors for the clarifications. I agree that FID may not be suitable in all scenarios. However, I strongly recommend including some form of quantitative analysis to demonstrate the effectiveness of the generated data—I'm glad to see this addressed in Figure 11. Regarding the loss function, I believe Section 3.1 of [Lou et al 2024](https://arxiv.org/pdf/2310.16834) provides a helpful illustration. Additionally, including a copy of the supplementary code demo could further strengthen the paper’s persuasiveness. Overall, I believe my initial score was positive, and I will keep it unchanged.

---

> > > ### Author Response · Authors · 2025-04-06
> > >
> > > Thank you for acknowledging our responses and for the reference to Lou et al. (2024); we will incorporate it into the revised version. The analyses suggested by you and the other reviewers have significantly strengthened the manuscript, and we’re grateful for the constructive feedback throughout.
> > >
> > > We also wanted to briefly follow up on your earlier concern regarding CelebA generation quality. During the rebuttal period, we managed to train a larger model, despite limited computational resources (just a single GPU), and obtained substantially improved generations which can be viewed [here](https://anonymous.4open.science/r/DSD-rebuttal-E860/figs/celebA_realizations_100.png). We’ll continue training and report updated FID scores in the final version, we believe that this can improve significantly with the allowed time.
> > >
> > > Additionally, we’re preparing a clean and complete code release with runnable examples. We hope this will not only support adoption in the scientific community but also inspire new applications in machine learning—such as budget-constrained inpainting and coherent colorization—beyond our original scope.
> > >
> > > We truly appreciate your positive review. Unfortunately, we didn’t receive input on our rebuttal from the other reviewers, but we’ve aimed to thoroughly address all raised points in our rebuttal and revision. We hope these efforts are taken into consideration in the final assessment.

---

### Official Review · Reviewer_EVW9 · 2025-03-19

**Overall Recommendation:** 2

**Summary:**

Discrete Spatial Diffusion (DSD) is a novel generative diffusion modeling framework specifically designed for discrete spatial domains, explicitly preserving mass throughout the diffusion processes. Traditional diffusion models typically assume continuous pixel intensities, thereby limiting their applicability to scientific datasets involving discrete, conserved physical quantities. DSD demonstrates its capability effectively in generating data for scientific tasks.

**Claims And Evidence:**

Quantitative performance comparisons (e.g., FID, Inception Score) against state-of-the-art diffusion or discrete generative models are missing.

**Essential References Not Discussed:**

It may related to : Planning with Diffusion for Flexible Behavior Synthesis

**Experimental Designs Or Analyses:**

Quantitative performance comparisons (e.g., FID, Inception Score) against state-of-the-art diffusion or discrete generative models are missing.

**Methods And Evaluation Criteria:**

In detailed comments

**Other Comments Or Suggestions:**

Background:

“The Markov process...” – I do not understand Figure 1. What are the meanings of variables such as x, y, I, and P(I_t>S | I_S)? They should be clearly defined. Also, clarify the context when mentioning "Gaussian" or "Discrete." Does this refer to the values of pixels?

“An alternative approach leverages Bayes... samples are exactly conditioned” – It appears similar to using a guidance function at inference time. The authors could refer to this paper [1].


Methods:

“where r is the transition rate of the particle jumping to one of their nearest neighbors” – I do not understand clearly what the transition rate means. Does it indicate how many particles jump to neighbors, or something else?

Figure 2(a): What is the timestep or step size for each image? Figure 2(b): What exactly is meant by "mass"?


Experiment:

Quantitative performance comparisons (e.g., FID, Inception Score) with other state-of-the-art diffusion or discrete generative models are missing.

[1] Planning with Diffusion for Flexible Behavior Synthesis

**Other Strengths And Weaknesses:**

Strength:

The idea to constrain particle numbers and only allow transfers to neighboring locations is interesting.

Weakness:

It’s difficult for me to understand the paper. Many terms are either not defined clearly or change names throughout the text.

There is no comparison with other models that do not strictly constrain particle numbers.

**Questions For Authors:**

Involving a guidance function during inference might achieve the same goal as DSD. Therefore, I think it would be beneficial to include experiments comparing the guidance function approach and DSD on scientific tasks.

**Relation To Broader Scientific Literature:**

NA

**Theoretical Claims:**

NA

---

> ### Author Rebuttal · Authors · 2025-04-01
>
> We thank the reviewer for their constructive feedback and questions.
>
>
> **1.1** [*Difficult to understand*]  We have tried our best to make it accessible and attempted to follow the conventions of [the theoretical paper](https://rb.gy/d7vqid). We will be happy to provide clarifications. It would be helpful if the reviewer can be more specific regarding confusing terms, missing definitions or explanations.
>
>
> **1.2** [*No comparison*] Our main point of the paper is to impose the hard constraint, strictly constraining the particle numbers. As such, the most important objective is to generate configurations which are exactly conditioned on the total intensity. In this sense, the rest of the models are failing the objective. The commonly adopted metrics, such as the FID score, are difficult to perform for exact conditioning for the approximate methods, as there is only a small set of images in the test set that has exactly a given total intensity, where FID between two given sets requires a large sample number.
>
>
> **1.3** [*Markov process*] We used a consistent notation as defined in the main text (line 156 of the text): “*We treat a digital image with discretized intensities $I_{x,y,c}$ at pixel $(x,y)$ in color channel $c$*”. $p(I_t \vert I_{t-1})$ are conditional distributions and a common notation used in the diffusion literature (e.g. Fig. 2, equation (1) in https://rb.gy/3v28z7). We have also provided a clear description of the continuous-state Gaussian generative diffusion models (line 41-48, left column), as well as state-of-the-art discrete-state generative models based on discrete-state Markov chains (line 72-98, left column), highlighting the difference to the Gaussian generative diffusion models in line 83-88, left column. We specified what the discreteness is referring to in line 147, right column: “*...Discrete Spatial Diffusion (DSD), noting the 'discreteness' refers to both the discretized intensity units and the discreteness of the spatial lattice... where the particles are allowed to reside.*"
>
>
> **1.4** [*Alternative approach*] Thank you. The guidance function in the provided paper makes sense for reinforcement-learning tasks - which the paper is about - where a cost function $J$ is naturally defined. For our generative tasks, it is not well-defined---in fact, it has to be learned either additionally as stated in our cited reference about *a posteriori* sampling, or it has to be built into the diffusion model and learned during training. This is because the "guidance function" is exact only when it is $\partial_{I_t} \log p(C\vert I_t)$ where $C$ is the condition. Here, $\log p(C\vert I_t)$ is not known.  While one can argue that we could try to impose heuristic guidance functions, our additional numerical experiments - which imposes a MSE between the generated and target total intensity - suggest that this is not a valid approach. We provide the Jupyter Notebooks (`/codes/Guided*.ipynb` in https://rb.gy/x1fjvy) generating the statistics of the total intensity of the generated samples with various guided functions (under `/figs/`). It is difficult to explain the experimental details with the 5K-character limit of the rebuttal, but we hope the notebooks are clear and we look forward to future discussion with the reviewer. In addition, we have implemented a conditional diffusion model taking the target total intensity as an additional input of the NN during training (`/codes/mass*.ipynb` in https://rb.gy/x1fjvy). The [results](https://rb.gy/ndpjo6) (see also **4.6**) suggest that (1) the total intensity can be *statistically* conditioned within the intensities in the data distributions, but not *exactly*, (2) after rounding and clipping the samples generated by Gaussian diffusion to `uint8`, a systematic additional bias is introduced, indicating that the conditioner is introducing physically impossible *negative intensities* to enforce the constraints, and (3) outside the data distribution, the conditioner completely fails. In comparison, DSD *always* has the required total intensity.
>
>
> **1.5** [*$r$*] The transition rate is a standard term referring to the transition probability per time. Cf. the cited textbooks, Van Kampen and Gardiner, or the published discrete-state generative model such as https://rb.gy/d7vqid and https://rb.gy/z7vfjv).
>
>
> **1.6** [*Timestep*] The diffusion model generating the schematic diagram has 1K steps and the snapshots were taken every 200 steps.
>
>
> **1.7** [*Mass*] We apologize. The mass stands for total particle numbers. We will revise.
>
>
> **1.8** [*Quant. performance*] Please refer to **1.2**, **1.4**, and **4.2**.
>
>
> **1.9** [*Additional exp.*] Please refer to **1.4**.
>
>
> **1.10** [*Summary*] We hope to have adequately addressed the points of confusion in the manuscript. We have also performed additional experiments following the suggestion to improve the work. These experiments help to show the benefits of our approach, and we thank the reviewer for the suggestions.

---

> > ### Comment · Reviewer_EVW9 · 2025-04-07
> >
> > I think the rebuttal is good and the Jupyter Notebooks is good. My previous rating was probably mostly based on the writing quality, but I may not have time to recheck the paper. If other reviewers continue to support accepting this paper, I would also agree with that decision.

---

> > > ### Author Response · Authors · 2025-04-08
> > >
> > > We sincerely thank the reviewer for taking the time to evaluate our manuscript and rebuttal. We are glad to hear that the reviewer found the new analysis satisfactory, and we are confident that the updated presentation now better supports a clearer and more accessible explanation of our contributions. These improvements will be further refined in the camera-ready version, should the paper be accepted. At this stage, the other two reviewers who have actively engaged in post-rebuttal discussions have expressed their strong support. We are encouraged by their recognition of the paper's novelty, analytical rigor, quantitative validations in the scientific datasets, and potential impact. If reviewer @EVW9 feels it is appropriate to support the manuscript, we kindly ask them to consider reflecting this in their score: currently it is still "weak rejection (but can be accepted)". This will help ensure that their support is visible during the AC's review and decision-making process, particularly in case this discussion does not receive close attention during the broader decision process. Once again, we truly appreciate the reviewer's time and thoughtful feedback.

---

### Decision · Program_Chairs · 2025-05-01

**Decision:**

Reject

**Comment:**

This paper proposes a diffusion model with exact mass conservation, which is required in certain scientific domains such as materials science. It demonstrates effectiveness across several domain-specific tasks as well as in image generation.

Reviewers acknowledge the novelty and relevance of the proposed approach for this specific setting, as well as the correctness of the derivation ensuring total intensity preservation in both the forward and backward processes. The method also shows general applicability across standard datasets (MNIST, CelebA). However, all reviewers raised a key concern: the lack of commonly used quantitative metrics in the image generation tasks makes it difficult to assess the model’s generative capabilities. Qualitatively, the image quality also falls short of the current state of the art. Additionally, there are concerns about the method’s scalability to large-scale, high-bit-depth images.

In the rebuttal, the authors provided the requested FID score but argued that such metrics are not appropriate for their constrained setting, where intensity mass must be conserved. During the discussion, reviewers agreed that FID is still the most appropriate metric for comparison under this setting. The concern of scalability is also addressed. While the method shows promise in domain-specific applications, its performance as a general image generative model remains behind recent methods, both qualitatively and quantitatively. Although the authors trained a larger model during the discussion period that showed improved results, the performance is still not strong enough to support the paper's claims. Further iteration is needed, and a resubmission would be more appropriate once model training is complete.